# Structural constraints on the emergence of oscillations in multi-population neural networks

**Jie Zang[1,2], Shenquan Liu[1]\*, Pascal Helson[2,3], Arvind Kumar[2,3]\***

[1]School of Mathematics, South China University of Technology, Guangzhou, China; [2]Division of Computational Science and Technology, School of Electrical Engineering and Computer Science, KTH Royal Institute of Technology, Stockholm, Sweden; [3]Science for Life Laboratory, Stockholm, Sweden

**\*For correspondence:**
mashqliu@scut.edu.cn (SL);
arvkumar@kth.se (AK)

**Competing interest:** The authors declare that no competing interests exist.

## eLife assessment

The present study offers **valuable** insights into the emergence of oscillations in neural networks. It underscores the importance of achieving a delicate balance between excitatory and inhibitory links, and deals with the topological conditions for oscillations. The study provides **solid** evidence in simple networks based on formal mathematical theory and advanced simulations, but the wider implications to biological networks would require a more detailed investigation into delays and nonlinearities.

**Abstract** Oscillations arise in many real-world systems and are associated with both functional and dysfunctional states. Whether a network can oscillate can be estimated if we know the strength of interaction between nodes. But in real-world networks (in particular in biological networks) it is usually not possible to know the exact connection weights. Therefore, it is important to determine the structural properties of a network necessary to generate oscillations. Here, we provide a proof that uses dynamical system theory to prove that an odd number of inhibitory nodes and strong enough connections are necessary to generate oscillations in a single cycle threshold-linear network. We illustrate these analytical results in a biologically plausible network with either firing-rate based or spiking neurons. Our work provides structural properties necessary to generate oscillations in a network. We use this knowledge to reconcile recent experimental findings about oscillations in basal ganglia with classical findings.

## Introduction

Oscillations are ubiquitous in dynamical systems *Strogatz, 2018*; *Pikovsky et al., 2002*. They have important functional consequences but can also cause system malfunction. In the brain for instance, oscillations take part in information transfer *Fries, 2015*; *Hahn et al., 2019*. However, persistent beta band (13-30 Hz) oscillations are associated with the pathological symptoms of Parkinson's disease (PD) *Brown et al., 2001*. Therefore, it is important to determine when and how a system of many interacting nodes (network) oscillates.

This question is usually very difficult to answer analytically. The main tool that can be used is the Poincaré–Bendixson theorem *Poincaré, 1880*; *Bendixson, 1901* which is only valid in two dimensions, which drastically reduces its applicability. In some cases, when we know the model parameters, it is possible to calculate whether the system will oscillate or not. However, often such parameters cannot be measured experimentally. For example, in most physical, chemical, and biological networks, it is

usually not possible to get the correct value of connectivity strength. By contrast, it is much easier to know whether two nodes in a system are physically connected and what is the sign (positive or negative) of their interactions. Therefore, it is much more useful to identify necessary structural conditions for the emergence of oscillations in a system without delays. A good example is the conjecture postulated by *Thomas, 1980* which relies on the sign of the loops present in the network. A loop (or a cycle) is said to be negative (resp. positive) when it has an odd (resp. even) number of inhibitory connections. According to Thomas' conjecture, when considering a coupled dynamical system ($\dot{x} = f(x)$ and $x(0) \in \mathbb{R}^n$) with a Jacobian matrix that has elements of fixed sign, the system can exhibit oscillations only if the directed graph obtained from the nodes' connectivity (Jacobian matrix) admits a negative loop of two or more nodes. This conjecture has been proven using graph theory for smooth functions $f$ *Snoussi, 1998*; *Gouzé, 1998*.

Thomas also conjectured that the assumption on the constant sign of the Jacobian matrix may not be necessary *Thomas, 2006*, that is having a negative loop in some domain of the phase space should be sufficient to generate oscillations. This condition is more realistic due to the ubiquity of the non-linearity in biological systems. For example, in the brain, even though neurons are (usually) either excitatory or inhibitory, the transfer function linking neurons is non-linear and can thus lead to elements of the Jacobian matrix having a non-constant sign. To the best of our knowledge, this last conjecture has not been proved yet but there are many examples of it. For instance, oscillations can emerge from a simple two populations, excitatory and inhibitory (EI), network Wilson-Cowan model (*Ledoux and Brunel, 2011*).

Here, we study the long-term behavior of the TLN model in the case of a single cycle interaction (without transmission delays). We show analytically that regardless of the sign of this loop, the system cannot oscillate when connections are too weak as the system possesses a unique globally asymptotically stable fixed point. However, when connections are strong enough (see Theorem 2), the system either possesses two asymptotically stable fixed points (positive loop) or a unique unstable fixed point (negative loop). In addition, the system can be shown to be bounded and thus, it has one of the following long term behavior: limit cycle, quasi-periodic or chaotic behavior. Interestingly, we can show that such dynamics can be shut down by introducing a positive external input to excited nodes.

We prove this conjecture for the threshold-linear network (TLN) model *Hartline and Ratliff, 1958* without delays which can closely capture the dynamics of neural populations. Therefore, it is implicit that our results do not hold at the neuronal level but rather at the level of neuron populations/brain regions for example the basal ganglia (BG) network which can be described a network of different nuclei.

In fact, we use our analytical results to explain recent experimental findings about the emergence of oscillations in the BG during PD. To this end, we used simulations of BG network models with either firing rate-based or spiking neurons. Within the framework of the odd-cycle theory, distinct nuclei are associated with either excitatory or inhibitory nodes. Traditionally, the subthalamic nucleus and globus pallidus (STN-GPe) subnetwork is considered to be the key network underlying the emergence of oscillations in PD (*Plenz and Kitai, 1996*; *Terman et al., 2002*; *Kumar et al., 2011*). However, recent experiments have shown that near complete inhibition of GPe but not of STN is sufficient to quench oscillations (*Crompe et al., 2020*). This observation contradicts several previous models and even clinical observations in which surgical removal of STN is used to alleviate PD symptoms. Our theory suggests that there are at least six possible cycles in the Cortex-BG network that have the potential to drive oscillations based on the connectivity structure. We show that even if STN is inhibited, other cycles can sustain pathological oscillations. Interestingly, we found that GPe is involved in five out of six oscillatory cycles and therefore GPe inhibition is likely to affect PD-related oscillations in most cases.

## Results

We study how the emergence of oscillations in a network of excitatory and inhibitory populations depends on the connectivity structure. We first consider a network of nodes with dynamics representing the average firing rate of a population. We derive structural conditions for the emergence of oscillations when the dynamics of individual nodes are described according to the threshold-linear network (TLN) model. Next, we use numerical simulations to test whether such results might still hold

on two other models: the Wilson-Cowan population rate-based model (*Wilson and Cowan, 1972*) and a network model of the BG with spiking neurons (see Methods).

## Structural conditions to generate oscillations

### Intuition behind the analytical results

There exist many ways to generate oscillations in a network. Oscillation can arise from individual nodes due to their intrinsic dynamics (e.g. a spiking neuron can exhibit a periodic behavior given its ionic channel composition *Lee et al., 2018*) or from the weights' dynamics when considering synaptic plasticity *Izhikevich and Edelman, 2008* or from transmission delays. Here, we assume that the system's ability to oscillate only depends on the connectivity structure: the presence of positive or negative loops and the connections strength (Jacobian matrix) within them. That is, we ignore node properties, weight dynamics and transmission delays. The main reason we ignore these important properties is that we want to know the structural properties that can induce oscillations – this question can only be asked when all other potential sources of oscillation are removed. Given the importance of delays in biological network such as BG, we will consider them in the simulations.

In this following, we give simple examples of the possibility of oscillation (or not) based on the connectivity characteristics of small networks without delays. Let us start with a network of two nodes. If we connect them mutually with excitatory synapses, intuitively we can say that the two-population network will not oscillate. Instead, the two populations will synchronize. The degree of synchrony will, of course, depend on the external input and the strength of mutual connections. If both these nodes are inhibitory, one of the nodes will emerge as a winner and the other will be suppressed *Ermentrout, 1992*. Hence, a network of two mutually connected inhibitory populations cannot oscillate either. We can extend this argument to three population networks with three connections that form a closed loop (or cycle; see top of *Figure 1a*). When all three connections in the cycle are excitatory, the three populations will synchronize. Essentially, we will have a single population. Thus, these two and three population motifs are not capable of oscillations.

The simplest network motif which is capable of oscillating thus consists of two mutually connected nodes: one excitatory and one inhibitory (EI motif: *Figure 1a*, bottom; *Ledoux and Brunel, 2011*). When there are three populations connected with three connections to form a cycle, the potential to oscillate depends on the number of inhibitory connections. A cycle with one inhibitory connection (EEI motif) can be effectively reduced to an EI motif and can therefore oscillate. However, when there are two inhibitory connections (EII motif, *Figure 1a*, top), the two inhibitory neurons engage in a winner-take-all type dynamics and the network is not capable of oscillations. Finally, if there are three inhibitory connections (i.e. all three nodes are inhibitory, III motif) the network enters in a winner-less-competition *Rabinovich et al., 2001* and can exhibit oscillations (*Figure 1a*, bottom).

These examples of two or three nodes suggest that a network can generate oscillations if there are one or three inhibitory connections in the network. We can generalize these results to cycles of any size, categorizing them into two types based on the count of their inhibitory connections in one direction (referred to as the odd cycle rule, as illustrated in *Figure 1b*). More complex networks can also be decomposed into cycles of size 2 N (where N is number of nodes), and predict the ability of the network to oscillate (as shown in *Figure 1c*).

These observations form the basis for the conjecture of *Thomas, 1980* that gives a necessary condition for oscillations to emerge. This condition is of course not sufficient. In the following we find additional constraints (input and minimum connection strength) needed to determine the emergence of oscillations in a network. To this end, we use the TLN model which captures the neural population dynamics to a great extent. After proving the key theorems, we test with simulation whether similar results hold on a more realistic Wilson-Cowan model and a model of BG with spiking neurons.

### Threshold linear network model

We consider the TLN$(W, b)$ in which individual nodes follow the dynamics

$$\frac{dx_i}{dt} = -x_i + \left[\sum_{j=1}^{n} W_{ij}x_j + b_i\right]_+, \quad i = 1, \ldots, n \tag{1}$$

where $n$ is the number of nodes, $x_i(t)$ is the activity level of the $i$th node at time $t \geq 0$, $W_{ij}$ is the connection strength from node $j$ to node $i$ and $[\cdot]_+ \overset{\text{def}}{=} \max\{\cdot, 0\}$ is the threshold non-linearity. For

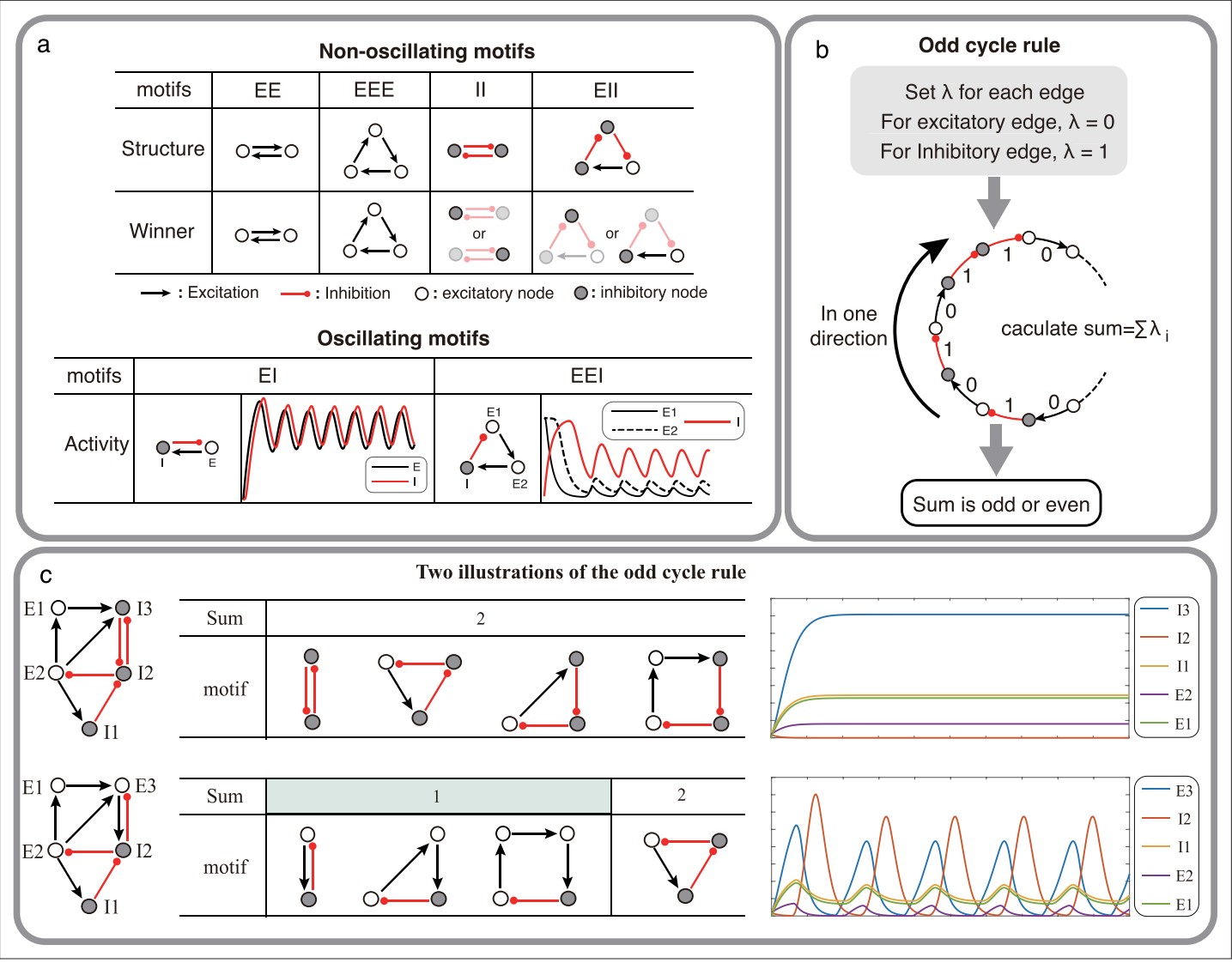

**Figure 1.** Structural condition for oscillations: odd inhibitory cycle rule and its illustrations. (**a**) Examples of oscillating motifs and non-oscillating motifs in Wilson-Cowan model. Motifs that cannot oscillate show features of Winner-take-all: the winner will inhibit other nodes with a high activity level. Inversely, the oscillatory ones all show features of winner-less competition, which may contribute to oscillation. (**b**) The odd inhibitory cycle rule for oscillation prediction with the sign condition of a network. (**c**) Illustrations of oscillation in complex networks. Based on the odd inhibitory cycle rule, Network I can't oscillate, while Network II could oscillate by calculating the sums of their motifs. The red or black arrows indicate inhibition or excitation, respectively. Hollow nodes and solid nodes represent excitatory and inhibitory nodes, respectively.

all $i \in [n] \stackrel{\text{def}}{=} \{1, \ldots, n\}$, the external inputs $b_i \in \mathbb{R}$ are assumed to be constant in time. We refer to a $n$ neurons network with dynamics given by *Equation 1* as TLN($W, b$).

In order to help the definition of cycle connectivity matrices, we define

$$C_n \stackrel{\text{def}}{=} \{(i,j) \in [n]^2 \mid i - j = 1\} \cup \{(1,n)\}.$$

We denote by $\delta_{i,I}$ the Kronecker delta which equals 1 when node $i$ is inhibitory and 0 otherwise (node $i$ is excitatory). In the following, we use the convention that node 0 is node $n$ and node $n + 1$ is node 1. For a given set of elements $\{y_k\}_{k \in \mathbb{N}}$ in $\mathbb{R}$, we will use the convention:

$$\prod_{k=i}^{j} y_k = 1 \text{ when } j < i. \tag{2}$$

We define $\mathcal{A} = \{a_1, \ldots, a_{n_I}\}$ as the ensemble of inhibited nodes ($\{k \in [n] \mid \delta_{k-1,I} = 1\}$) put in order such that $a_1 < \ldots < a_{n_I}$. Denoting by $\mathrm{card}(\cdot)$ the cardinal function, we have that $\mathrm{card}(a) = n_I$. We also use the cycle convention for $\mathcal{A} : a_{n_I+1} = a_1$.

## Analytical results

### Theorem 1

Let a network of inhibitory and excitatory nodes be connected through a graph $G$ which does not contain any directed cycle. Assume that its nodes follow TLN($W, b$) dynamics (**Equation 1**) with

$$\mathrm{W}_{ij} = \begin{cases} w_{ij}(-1)^{\delta_{j,I}} & when\ edge\ i \leftarrow j \in G \\ 0 & otherwise, \end{cases}$$

where $w_{ij} \in \mathbb{R}^+ w_{ij} \in \mathbb{R}^+$

Then, TLN($W, b$) has a unique globally asymptotically stable fixed point.

### Theorem 2

Let $G$ be a cyclical graph with $n_I \in \mathbb{N}^+$ inhibitory nodes and $n_E \in \mathbb{N}$ excitatory nodes such that $n_I + n_E \geq 2$ ($\geq 3$ when $n_I = 1$). Assume that the nodes follow the TLN($W, b$) dynamics (**Equation 1**) with for all $i, j \in [n]$, $w_j \in \mathbb{R}^+$,

$$\mathrm{W}_{ij} = \begin{cases} w_j(-1)^{\delta_{j,I}} & when\ (i,j) \in C_n \\ 0 & otherwise, \end{cases}$$

and $b_i = 0$ when the node $i - 1$ is excitatory and $b_i > 0$ otherwise. Moreover, using convention (**Equation 2**), assume that the initial state is bounded,

$$\forall j \in \{a_k, \ldots, a_{k+1} - 1\}, \quad x_j(0) \in [0, b_{a_k} \prod_{i=a_k}^{j-1} w_i]. \tag{3}$$

Then, the long-term behavior of the network depends on the following conditions,

$$\forall k \in [n_I], \quad \prod_{i=a_k}^{a_{k+1}-1} w_i < \frac{b_{a_{k+1}}}{b_{a_k}}, \tag{4}$$

$$\prod_{i=a_k}^{a_{k+1}-1} w_i > \frac{b_{a_{k+1}}}{b_{a_k}}, \tag{5}$$

$$\sqrt[n]{\prod_{i=1}^{n} w_i} < \frac{1}{cos(\pi/n)}, \tag{6}$$

$$\sqrt[n]{\prod_{i=1}^{n} w_i} > \frac{1}{cos(\pi/n)}, \tag{7}$$

If $n_I$ is even and

- **Equation 4** is satisfied, TLN($W, b$) has a unique globally asymptotically stable fixed point with support $[n]$,
- **Equation 5** is satisfied, TLN($W, b$) has two asymptotically stable fixed points with strict complementary subsets of $[n]$ as supports.

If $n_I$ is odd and

- **Equation 4** is satisfied, TLN($W, b$) has a unique fixed point which is globally asymptotically stable and its support is $[n]$,

- **Equation 5** and **Equation 6** is satisfied, TLN($W, b$) has a unique fixed point which is asymptotically stable (not globally) and its support is $[n]$,
- **Equation 5** and **Equation 7** are satisfied, TLN($W, b$) has a unique fixed point which is unstable and has $[n]$ as support.

## Remark 1

First, note that **Equation 4** implies

$$\sqrt[n]{\prod_{i=1}^{n} w_i} < 1, \tag{8}$$

and similarly, **Equation 5** implies

$$\sqrt[n]{\prod_{i=1}^{n} w_i} > 1. \tag{9}$$

In addition, the bound on the initial state **Equation 3** can be easily removed. We use it because it eases the proof as we then don't need to introduce technical details that are not interesting for this study.

Then, Theorem 2 says that a possible condition for the one cycle TLN to oscillate is that the number of inhibitory nodes is odd when the connection strength are strong enough (i.e. **Equation 5** and **Equation 7**). In that case, the system has no stable fixed point and from Lemma 1 it is bounded so it has one of limit cycle, quasi-periodic or chaotic behaviors. In particular, Theorem 2 states that the odd number of inhibitory nodes is not sufficient. Indeed, when **Equation 4** holds and $n_I$ is odd, no oscillations are possible as the fixed point is globally stable. It is also the case when $n_I$ is even which corresponds to Thomas' conjecture.

Finally, there is a gap in between the conditions (between **Equation 4** and **Equation 5** for example) for which the long-term behavior is not determined.

## Remark 2

In particular, if for all $i \in [n]$, $w_i = w \in \mathbb{R}_+^*$ and for all $k \in [n_I]$, $b_{a_k} = b \in \mathbb{R}_+^*$, then the dynamics of the system only depends on $w$. When $n_I$ is even: $w < 1$ implies that TLN($W, b$) have a unique globally asymptotically stable fixed point; $w > 1$ implies that the fixed point for $w < 1$ becomes unstable and TLN($W, b$) has two more asymptotically stable fixed point. If $n_I$ is odd, TLN($W, b$) only has a unique fixed point which is asymptotically stable when $w < \frac{1}{cos(\pi/n)}$ (globally when $w < 1$) and unstable when $w > \frac{1}{cos(\pi/n)}$.

## Remark 3

In Theorem 2, we assume that the external inputs are absent for excited nodes. Assume that the external input to any excited node, say node $a_k < i < a_{k+1}$, is strictly positive. Then, bounding its dynamics as in Lemma 1, we know that its activity will be more than $b_i$. Hence, the next inhibited node $a_{k+1}$ can be silenced forever if

$$b_i \prod_{j=i}^{a_{k+1}-1} w_j > b_{a_{k+1}},$$

thus destroying the cycle structure and thus preventing oscillation from emerging.

On the other hand, if the external inputs to excited nodes are strictly negative, Theorem 2 conclusion will be similar but now with condition described in **Equation 4** replaced by

$$b_{a_k} \prod_{i=a_k}^{a_{k+1}-1} w_i - \sum_{j=a_k+1}^{a_{k+1}-1} b_j \prod_{i=j}^{a_{k+1}-1} w_i < b_{a_{k+1}}.$$

This means that cycles with even (odd) inhibitory nodes need strong enough connections to generate multi-stability (limit cycle). We now clarify that the latter condition relates to weights' strength. With $w = (w_1, \cdots, w_n)$ and using the set of functions increasing functions $(f_i)_{1 \le i \le n}$ such that

$$f_i^{w,b}(x) = w_{i-1}x_{i-1} - b_i$$

one can write the last condition as

$$f_{a_{k+1}}^{w,b} \circ \cdots \circ f_{a_k}^{w,b}(b_{a_k}) < 0.$$

Hence, the left term is increasing with any weight strength.

One should also note that under this negative input assumption to excited nodes, when weights are weak, the support of the fixed point might be different from $[n]$. In particular, some excited nodes might not belong to the support.

### Remark 4

When the decay rates are not the same (here all of them are $-1$), similar results hold but then the conditions for stability are more difficult to state precisely. Finally, when considering the EI network (two nodes), the system always admits to a unique globally asymptotically stable fixed point with support $\{1, 2\}$. Indeed, it is easy to show that the system will always reach the domain where the inhibitory node is small enough so that one can remove the threshold function in *Equation 1* and thus the eigenvalues of the Jacobian matrix are $\pm i\sqrt{w_1 w_2} - 1$. No oscillations are then possible, which is an easy example to show that negative loops are not sufficient to generate oscillation in non smooth dynamical systems.

Similar results have been shown by *Gouzé, 1998*; *Snoussi, 1998*. Considering dynamical systems of the form $\dot{x} = f(x)$ where $f$ is a continuously differentiable function on a given open convex set and $f$ has a constant sign Jacobian matrix, they used graph theory methods to show that negative loop in this matrix is a necessary condition to generate oscillations. In our case, $f$ is not continuously differentiable, the Jacobian matrix elements can change sign within the state space and we show that there is a need of additional constraints for oscillations to arise. A formal proof of the aforementioned theorems is provided in Appendix A by using classical dynamical theory tools.

## Intuition behind the proof of the theorems

The idea behind our proof can be explained graphically. We assume that nodes cannot oscillate due to their intrinsic activity and a fixed external input only drives them to a non-zero activity which does not change over time. Therefore, they need input from their pre-synaptic (upstream) nodes to change their state in a periodic manner to generate oscillations. In such a network, if we perturb the node $i$ with a pulse-like input, it is necessary that the perturbation travels through the network and returns to the node $i$ with a 180° phase shift (i.e. with an inverted sign). Otherwise, the perturbation dies out and each node returns to a state imposed by its external input.

In a network without directed cycles, it is possible to sort the nodes into smaller groups where nodes do not connect to each other (*Figure 2a*). That is, a network with no directed cycles, can be rendered as a feed-forward network in which the network response by definition does not return to the node (or group) that was perturbed. Such a network can only oscillate when the intrinsic dynamics of individual nodes allow for oscillatory dynamics.

However, having a directed cycle is no guarantee of oscillations because network activity must return to the starting node with a 180° phase shift. This requirement puts a constraint on the number of inhibitory connections in the cycles. When we assume that there are no delays (or the delay is constant) in the connections, excitatory connections do not introduce any phase shift, however, inhibitory connections shift the phase by 180° (in the simplest case invert the sign of the perturbation). Given this, when a cycle has an even number of inhibitory connections the cycle cannot exhibit oscillations (*Figure 2b*,top). However, replacing an inhibitory connection by an excitatory one can render this cycle with an ability to oscillate (*Figure 2b*, bottom). Therefore, odd number of inhibitory appears to be necessary for oscillation to emerge.

## The effect of network parameters on oscillations

To test the validity of our theorems in more realistic biological neuronal networks, we numerically simulated the dynamics of the Wilson-Cowan model. Specifically, we investigated the role of synaptic transmission delays, synaptic weights, external inputs and self-connection in shaping the oscillations when the network has directed cycles. In particular, we focused on two networks: the III motif with

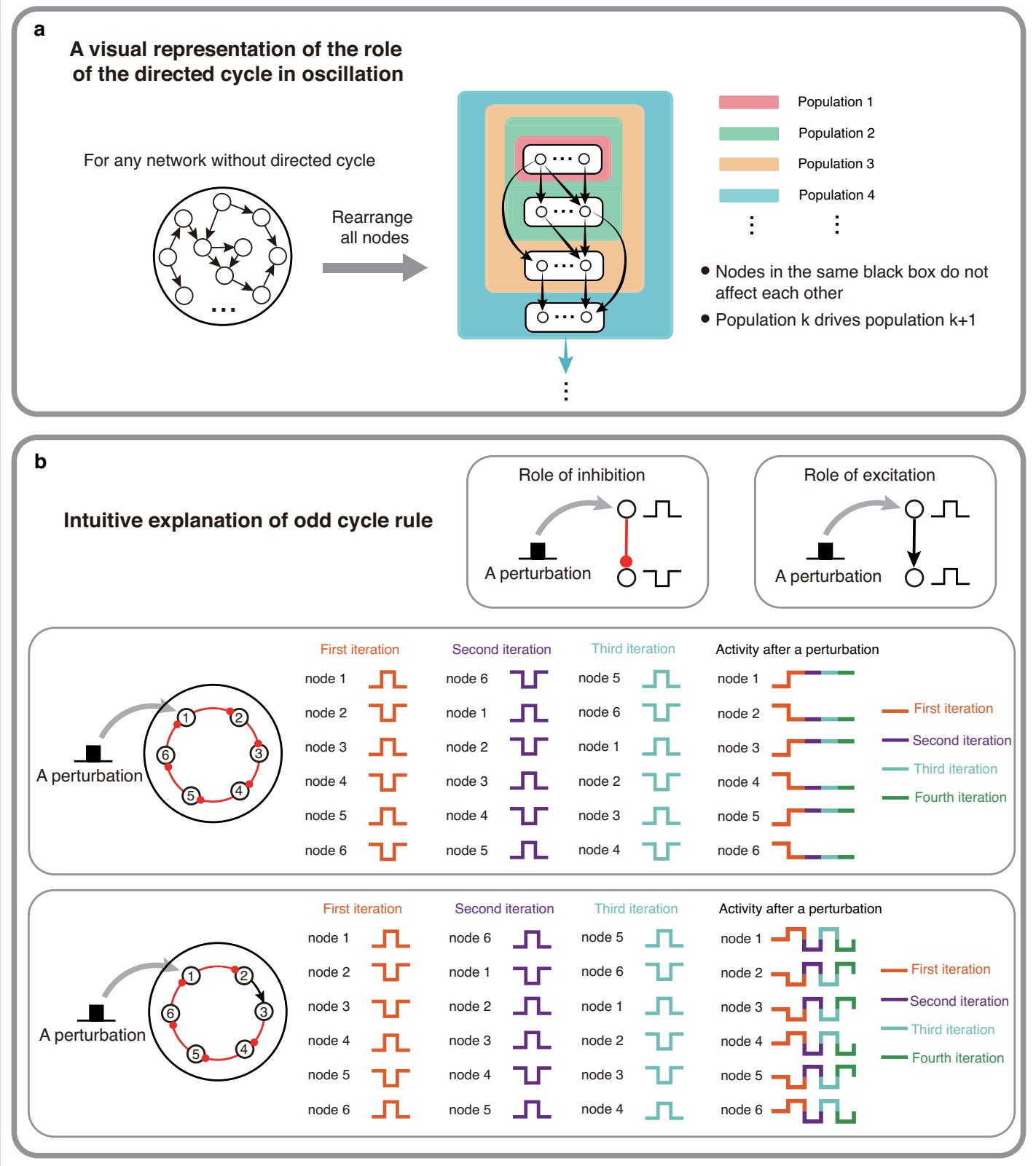

**Figure 2.** The intuitive explanations of Theorem 1 and 2. (**a**) A visual representation of why directed cycles are important in network oscillation. By rearranging all nodes, any network without directed cycles can be seen as a feed-forward network which will make the system reach a stable fixed point. (**b**) An intuitive explanation of the odd inhibitory cycle rule by showing the activities of two 6-node-loops. Odd inhibitory connections (bottom) can help the system oscillate, while even inhibitory connections has the opposite effect.

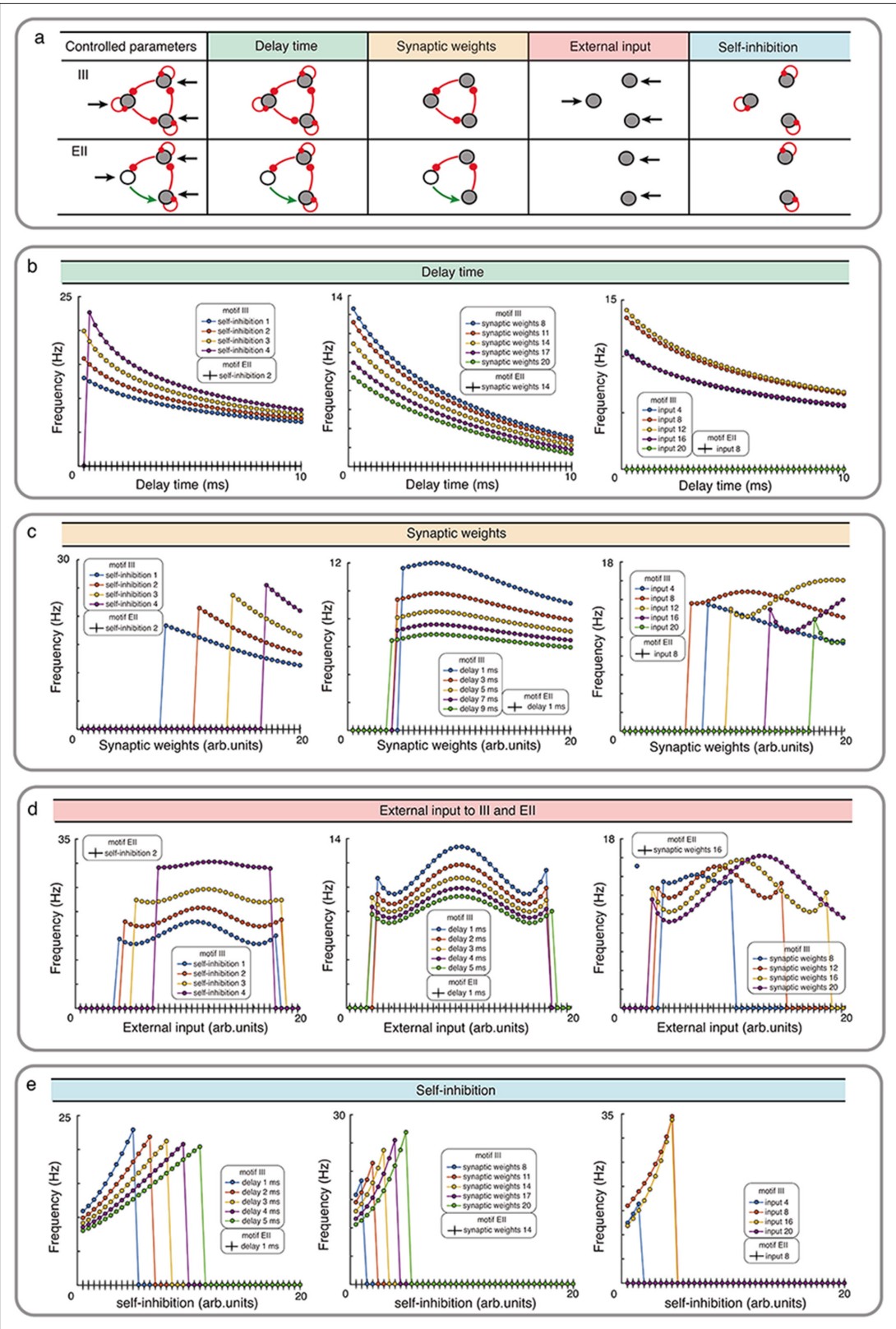

**Figure 3.** Influence of network properties on the oscillation frequency in motifs III and EII with Wilson-Cowan model. (**a**) The changed network parameters are shown in the table. Red (green) connections are inhibitory (excitatory) and black arrows are the external inputs. (**b-e**) We systematically varied the synaptic delay time **b**, synaptic weights **c**, external input **d**, and self-connection **e**. These parameters were varied simultaneously for all the synapses i.e. in each simulation all synapses were homogeneous. Green, orange, red and turquoise respectively show the effect of synaptic delay,

*Figure 3 continued on next page*

three inhibitory nodes (odd inhibitory links) and the EII motif with one excitatory and two inhibitory nodes (even inhibitory links).

Our numerical simulation showed that for a wide range of parameters (synaptic delays, synaptic weights, external input and self-inhibition), while III network showed oscillations, EII network only resulted in transient oscillations (*Figure 3*, *Figure 3—figure supplements 1 and 2*). The oscillation frequency however depended on the exact value of the synaptic delays, synaptic weights and external inputs. For instance, increasing the synaptic delay reduced the oscillation frequency (*Figure 3b*). Synaptic delays play a more important role in shaping the oscillations in an EI type network (see *Figure 3—figure supplement 3*, *Appendix 2—table 1*). In networks with even number of inhibitory connections (e.g. EII, EEE, II), synaptic delays are the sole mechanism for initiating oscillations, however, unless delays are precisely tuned such oscillations will remain be transient (see *Figure 3—figure supplement 2*). The effect of increasing the synaptic strength was contingent on the external inputs. In general, increasing the synaptic strength resulted in a reduction in the oscillation frequency (*Figure 3c*). Next, oscillation frequency changed in a non-monotonic fashion as a function of external input irrespective of the choice of other parameters (*Figure 3d*). Typically, a mid-range input strength resulted in maximum oscillation frequency. Finally, increasing the self-connection of nodes increased the oscillation frequency but beyond a certain self-connection the node was completely silenced and it changed the network topology and oscillations disappeared (*Figure 3e*).

Overall, these results are consistent with our rule that odd number of inhibitory nodes and strong enough connections are necessary to induce oscillations in a directed cycle. The actual frequency of oscillations depends on specific network parameters.

## Oscillators in the cortex-basal ganglia network

Next, we use our theorem to explain recent experimental observations about the mechanisms underlying the emergence of oscillations in the BG. Emergence of 15–30 Hz (beta band) oscillations in the cortico-basal ganglia (CBG) network is an ubiquitous feature of PD *Raz et al., 2000*; *Bergman et al., 1998*; *Sharott et al., 2014*; *Neumann et al., 2016*. Based on their connectivity and activity subthalamic nucleus (STN) and the globus pallidus externa (GPe) subnetwork has emerged as the most likely generator of beta oscillations *Plenz and Kital, 1999*; *Bevan et al., 2002*. The STN-GPe subnetwork becomes oscillatory when their mutual connectivity is altered *Terman et al., 2002*; *Holgado et al., 2010* or neurons become bursty *Tachibana et al., 2011*; *Bahuguna et al., 2020* or striatal inputs to GPe increase *Kumar et al., 2011*; *Mirzaei et al., 2017*; *Sharott et al., 2017*; *Chakravarty et al., 2022*. However, oscillations might also be generated by the striatum *McCarthy et al., 2011*, by the interaction between the direct and hyperdirect pathways *Leblois et al., 2006* and even by cortical networks that project to the BG (*Brittain and Brown, 2014*). Recently, *Crompe et al., 2020* used optogenetic manipulations to shed light on the mechanisms underlying oscillation generation in PD. They showed that GPe is essential to generate beta band oscillations while motor cortex and STN are not. These experiments force us to rethink the mechanisms by which beta band oscillations are generated in the CBG network.

To better understand when GPe and/or STN are essential for beta band oscillations, we identified the network motifs which fulfill the odd inhibitory cycle rule. For this analysis, we excluded D1 SPNs because they have a very low firing rate in the PD condition *Sharott et al., 2017*. In addition, cortex is assumed as a single node in the CBG network.

The CBG network can be partitioned into 246 subnetworks with 2, 3, 4, 5, 6, or 7 nodes (see *Figure 4—figure supplements 1–6*). Among these partitions, there are five loops (or cycles) in the CBG network with one or three inhibitory projections: Proto-STN, STN-GPi-Th-cortex, STN-Arky-D2-Proto, Proto-Arky-D2, Proto-FSN-D2, and Proto-GPi-Th-Cortex-D2 (*Figure 4a*). One or more of these 6 loops appeared in 96 (out of 246) subnetworks of CBG (see *Figure 4b*, colors indicate different loops). Larger subnetworks consisting of 5 and 6 nodes have multiple smaller subnetworks (with 2 or 3 nodes) that can generate oscillations (boxes with multiple colors in *Figure 4b*).

Based on our odd inhibitory cycles in BG, we found three oscillatory subnetworks which do not involve the STN (*Figure 4a*, cyan, green and purple subnetworks). However, each of these oscillatory subnetworks involves Prototypical neurons (from the GPe) which receive excitatory input from STN. Therefore, it is not clear whether inhibition of STN can affect oscillations or not. To address this question, we first simulated the dynamics of a four-node motif (*Figure 4c* top) using the Wilson-Cowan

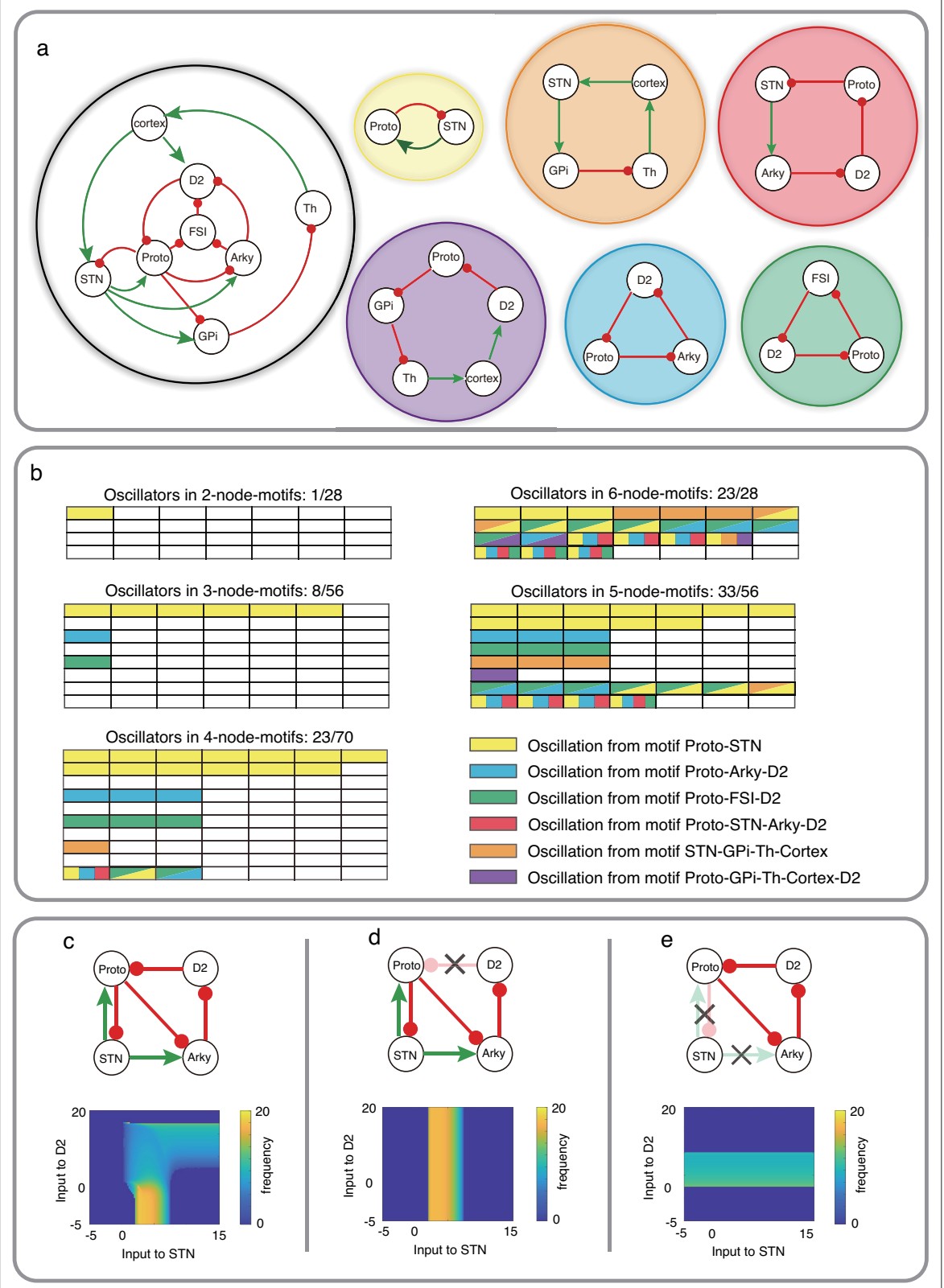

**Figure 4.** Schematic of CBG network model with potential oscillators and the interaction between two oscillators in Wilson-Cowan model. (**a**) CBG structure with red lines denoting inhibition and green lines denoting excitation, along with five potential oscillators based on the odd inhibitory cycle rule. (**b**) Oscillation in all BG motifs from 2 nodes to 6 nodes based on the odd inhibitory cycle rule. Each grid represents a separate motif. We use different colors to mark potential oscillators in each motif in BG, and each color means an oscillator from panel **a**. For more details, see *Figure 4—*

*Figure 4 continued on next page*

*Figure 4 continued*

**figure supplements 1–6**. (**c**) The reaction of oscillation frequency to different external inputs to D2 and STN in a BG subnetwork. External inputs to Proto and Arky are 1 and 3, respectively. (**d**) Same thing as c but ruining the connection from D2 to Proto. (**e**) Same thing as c but destroying the connections from STN and increasing the input to Proto from 1 to 4.

The online version of this article includes the following figure supplement(s) for figure 4:

**Figure supplement 1.** All 2-node-motifs in CBG network.

**Figure supplement 2.** All 3-node-motifs in CBG network.

**Figure supplement 3.** All 4-node-motifs in CBG network.

**Figure supplement 4.** All 5-node-motifs in CBG network.

**Figure supplement 5.** All 6-node-motifs in CBG network.

**Figure supplement 6.** All 7-node-motifs in CBG network.

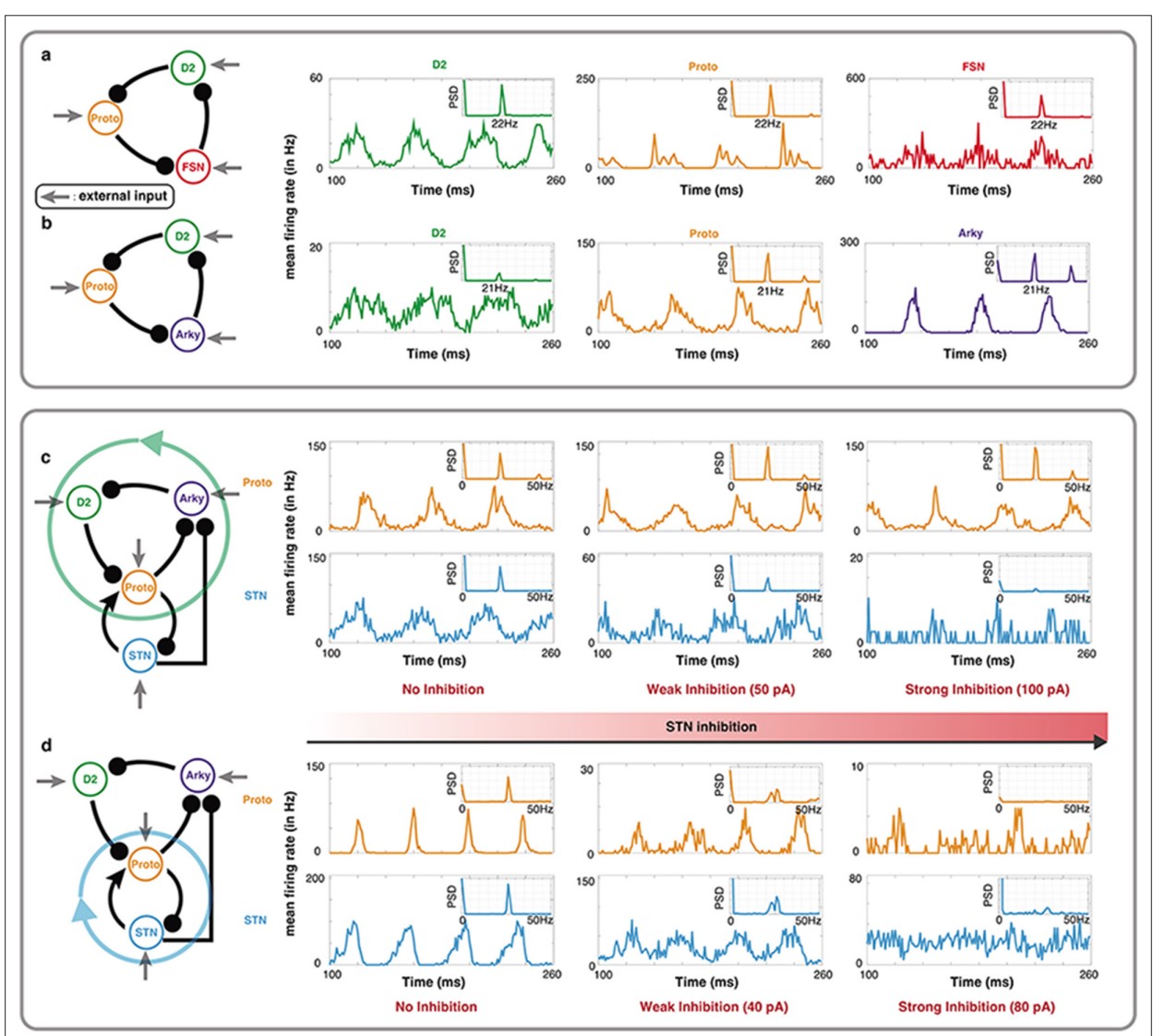

**Figure 5.** Oscillations in a leaky integrate-and-fire (LIF) spiking neuronal network model of specific BG motifs. (a-b) Average peristimulus time histograms (PSTH) of all neurons in a Proto-FSN-D2 and (b) Proto-Arky-D2 motifs under Parkinson condition with power spectral density (PSD) at the top right. (c) PSTH of Proto and STN in a BG subnetwork with motif Proto-Arky-D2 as the oscillator during different STN inhibition. (d) Same thing as (c) but changing the oscillator from Proto-Arky-D2 to Proto-STN.

*Figure 3 continued*

synaptic strength, external input and self-inhibition. See the *Figure 3—figure supplement 1* and *Figure 3—figure supplement 3* for more detailed results about III and EI network motifs.

The online version of this article includes the following figure supplement(s) for figure 3:

**Figure supplement 1.** Influence of III network properties on the oscillation frequency in Wilson-Cowan model.

**Figure supplement 2.** Effect of synaptic delays on motifs with even inhibition in the Wilson-Cowan model.

**Figure supplement 3.** Influence of EI network properties on the oscillation frequency in Wilson-Cowan model.

type model (see Methods). In this subnetwork, we have three cycles: Proto-STN loop with one inhibitory connection, Proto-STN-Arky-D2 loop with three inhibitory connections and Proto-Arky-D2 with three inhibitory connections.

We systematically varied external inputs to the STN and D2-SPNs and measured the frequency of oscillations (see Methods). We found that for weak inputs to the D2-SPNs, the Proto-STN subnetwork generated oscillations for weak positive input (*Figure 4c* bottom). However, as the input to D2-SPNs increased, the oscillation frequency decreased and oscillations were observed even for very strong drive to STN (*Figure 4c* bottom). That is, in this model, both Proto-STN and Proto-D2-Arky subnetworks compete for oscillations, which subnetwork wins depending on their inputs. To disentangle the oscillations of each of these two subnetworks, we performed 'lesion' experiments in our model (see Methods). These experiments also mimicked lesions performed in non-human primates *Tachibana et al., 2011*.

When we removed the D2-SPN to Proto projections, the network could oscillate but only because of the Proto-STN subnetwork (*Figure 4d*). In this setting, we get relatively high-frequency beta band oscillations but only for a small range of excitatory inputs to the STN (*Figure 4d*, bottom). In this setting, inhibition of STN would certainly abolish any oscillation. Next, we removed the STN output (equivalent to inhibition of STN), the Proto-D2-Arky subnetwork generated oscillations for weak positive inputs to the D2-SPNs (*Figure 4e*, bottom). Note that unlike in *Figure 4c*, here we injected additional input to Proto to compensate for the loss of excitatory input from STN and to ensure that it had sufficient baseline activity. The frequency of Proto-D2-Arky oscillations was smaller than that observed for the Proto-STN subnetwork because the former involves a three synapses loop. However, as we have shown earlier, frequency of oscillation can be changed by scaling the connection weights or external inputs (*Figure 3*). Overall, these results suggest that, in principle, it is possible for CBG network to oscillate even when STN is removed from the network.

## Oscillations in model of basal ganglia with spiking neurons

Thus far we have only illustrated the validity of our theorems in a firing rate-based model. To be of any practical value to brain science, it is important to check whether our theorems can also help in a network with spiking neurons. To this end, we simulated the two subnetworks with 3 inhibitory connections: Proto-D2-FSN and Proto-D2-Arky (see Methods). These subnetworks were simulated using a previous model of BG with spiking neurons *Chakravarty et al., 2022*.

The subnetworks Proto, Arky, D2-SPN, FSN have very little recurrent connectivity to oscillate on their own. We provided Poisson type external input. All neurons in a subnetwork received the same input rate but a different realization of the Poisson process. Both Proto-D2-FSN (*Figure 5a*) and Proto-D2-Arky (*Figure 5b*) subnetworks showed $\beta$-band oscillations. In the loop Proto-D2-FSN, D2-SPN neurons have a relatively high firing rate. This could be a criterion to exclude this loop as a potential contributor to the beta oscillations.

Next, we mimicked the STN inhibition experiments performed by *Crompe et al., 2020* in our model. To this end, we simulated the dynamics of BG network excluding D1-SPNs (because of their low firing rate in PD condition) and FSN (because with FSNs in the oscillation loop, D2-SPNs may have non-physiological firing rates). In this reduced model of BG, we changed inputs to operate in a mode where either Proto-D2-Arky (*Figure 5c*) or Proto-STN (*Figure 5d*) loop was generating the oscillations. In both cases, we systematically increased the inhibition of STN neurons.

In the Proto-D2-Arky mode, as we inhibited STN neurons, firing rate of the Proto neurons decreased and oscillations in STN population diminished but Proto neurons showed clear beta band oscillations (*Figure 5c*). By contrast, and as expected when STN-Proto loop was generating the oscillations,

increasing the STN inhibition abolished the oscillations in both STN and Proto neurons (*Figure 5d*). In STN-Proto loop, when STN is inhibited, there is no cycle left in the network and therefore oscillations diminished, whereas the Proto-D2-Arky loop remains unaffected by the STN inhibition (except for a change in the firing rate of the Proto neurons). As shown in *Figure 4c*, whether oscillations are generated by the Proto-D2-Arky or STN-Proto loop depends on the relative input to the D2 or STN neurons. So it is possible that in rodents, D2-SPN have stronger input from the cortex than STN and therefore, oscillations survive despite near complete inhibition of STN.

## Discussion

Here, we prove in a single cycle TLN model and illustrate with numerical simulations of biological networks that when the number of inhibitory nodes in a directed cycle is odd and connections are strong enough, then the system has the potential to oscillate. In 1981, *Thomas, 1980* conjectured that at least one negative feedback loop (i.e. a loop with an odd number of repressors) is needed for gene regulatory networks to have periodic oscillating behavior. This conjecture was proven for Boolean dynamical systems by *Snoussi, 1998* and *Gouzé, 1998*. But their proof required that node transfer-function is differentiable everywhere. We here prove a more complete theorem for a case where node transfer-function is threshold-linear as is the case for many network in the brain. Thus, together with previous results of *Snoussi, 1998* and *Gouzé, 1998* we further expand the scope within which we can comment on the potential of a network to generate oscillation based only on the connectivity structure alone. In addition, we complete this condition by one on weights' strength stating that the latter needs to be strong enough for the system to possibly oscillate. Eventually, oscillations can be quenched by adding positive external input to excited nodes.

A key assumption of our analysis is that there are no delays in the network. Indeed, delays within and between subnetwork connections can have a big effect on the oscillations *Kim et al., 2020*. In the numerical simulations of BG network, we included biologically realistic synaptic delays (i.e. connection delays were shorter than the time constants of the neurons). Our results suggest that such delays do not influence our results and they only determine the oscillation frequency. But it is not possible to comment on how the results may change when delays become longer than the time constant of the node.

### Interactions between input and network structure

Previous models suggest that when we excite the excitatory node or inhibit the inhibitory node oscillations can emerge and strengthen *Kumar et al., 2011*; *Ledoux and Brunel, 2011*. By contrast, when we inhibit the excitatory node or excite the inhibitory node, oscillations are quenched. This can be summarised as the 'Oscillations Sign Rule'. Let us label the excitatory population as positive and inhibitory as negative. Let us also label excitatory inputs as positive and inhibitory inputs as negative. Now if we multiple the sign of the node and sign of the stimulation, we can comment on the fate of oscillations in a qualitative manner. For example, inhibition of inhibitory nodes would be $- \times - = +$ that is oscillations should be increased and when we inhibit excitatory nodes, it would be $- \times + = -$ that is oscillations should be decreased. The 'Oscillation sign rule' scales to larger network with more nodes. With the 'Odd Cycle Rule' as we have shown we can comment on whether a directed cycle will oscillate or not from the count of inhibitory links. When we combine the 'Oscillations Sign Rule' with the 'Odd Cycle Rule' we can get a more complete qualitative picture of whether a stimulating a node in a network will generate oscillations or not.

### Interaction between node properties and network structure

In our proof we have assumed that nodes follow rather simple dynamics and have a threshold-linear transfer-function. In reality nodes in physical, chemical and biological systems can have more complex dynamics. For instance, biological neurons have the property of spike frequency adaptation or rebound spiking. Similarly, synapses in the brain can increase or decrease their weights based on the recent history of inputs which is referred to as the short-term-facilitation or short-term-depression of synapses *Stevens and Wang, 1995*. Such biological properties can be absorbed in the network structure in the form of an extra inhibitory or excitatory connection. When nodes can oscillate given their

intrinsic dynamics then the question becomes more about whether a network structure can propagate oscillations to other nodes.

## Oscillations in the basal ganglia

We applied our results to understand the mechanisms underlying the emergence of PD-related pathological oscillations in the BG. Given that there are 8 key neuron populations in the BG, we enumerated 238 possible directed cycles. From 2-node-motifs to 6-node-motifs, our odd cycle rule identified 88 potential directed cycles that can generate oscillations. Among these, 81 cycles feature GPe (either Proto or Arky type or both) and 66 feature STN. Which specific cycle underlies oscillations depends on the exact input structure. For instance, when input to STN is higher than the D2 neurons, the STN-GPe network generates oscillations. But when inputs to D2 neurons are stronger, the D2-Proto-Arky cycle can become the oscillator. That is, STN is not necessary to generate oscillations in the BG. Our results also suggest that besides focusing on the network connectivity, we should also estimate the inputs to different nodes in order to pinpoint the key nodes underlying the PD-related pathological oscillations - that would be the way to reconcile the recent findings of *Crompe et al., 2020* with previous results.

We would like to note that recently, *Ortone et al., 2023* and *Azizpour Lindi et al., 2023* have provided a more quantitative explanation of the experimental observation by *Crompe et al., 2020*. They also eventually resort to the D2-Proto-Arky cycle to explain BG oscillations in the absence of STN. Our work provides the necessary theory to explain their results.

## Beyond neural networks

In this work, we have used the odd cycle rule to study oscillations in BG. However, oscillatory dynamics and the odd cycle rule show up in many chemical, biological and even social systems such as neuronal networks *Bel et al., 2021*, psychological networks *Greenwald et al., 2002*, social and political networks *Leskovec et al., 2010*; *Milo et al., 2004*; *Heider, 1946*; *Cartwright and Harary, 1956*, resting-state networks in autism *Moradimanesh et al., 2021* and gene networks *Farcot and Gouzé, 2010*; *Allahyari et al., 2022*. In fact, originally Thomas' conjecture *Thomas, 1980* about the structural conditions for oscillations was made for gene regulatory networks. Therefore, we think that insights obtained from our analytical work can be extended to many other chemical, biological and social networks. It would be interesting to check to what extent our prediction of quenching oscillation by exciting the excitatory nodes holds in other systems besides biological neuronal networks.

# Methods

To study the emergence of oscillations in the BG, we used three models: Threshold-Linear Network (TLN), Wilson-Cowan model and network with spiking neurons. TLN model (*Equation 1*) was used here to rigorously prove that simple conditions, such as the odd inhibitory cycle rule, can lead to oscillations (Theorems 1 and 2). Wilson-Cowan type firing rate-based model was used to find the structural constraints on oscillations and to determine the effect of network properties (such as delays, synaptic weights, external inputs, and self-inhibition) on the emergence of oscillations. Finally, to demonstrate the validity of the odd inhibitory cycle rule in a more realistic model, we use a network with spiking neurons.

## Wilson-Cowan dynamics

In the firing rate-based models, we reduced each CBG subnetwork to a single node. To describe firing rate dynamics of such a node, we used the classic Wilson-Cowan model *Wilson and Cowan, 1972*

$$\tau \frac{dr_i(t)}{dt} = -r_i(t) + F\left(\sum_{j=1}^{n} w_{ij}r_j + I_i^{ext}\right) \tag{10}$$

where $r_i(t)$ is the firing rate of the $i$th node, $\tau$ is the time constant of the population activity, $n$ is the number of nodes (or subnetworks), $w_{ij}$ is the strength of connection from node $j$ to $i$, and $I_i^{ext}$ is the external input to the population. $F$ is a nonlinear activation function relating output firing rate to input, given by

**Table 1.** Parameters of III network for *Figure 3*, *Figure 3—figure supplement 1*.

| Populations | Synaptic weights | | | External input | Delay |
|---|---|---|---|---|---|
| | I1 | I2 | I3 | | |
| I1 | 0(−20−0) | 0 | −15(−20−0) | 6(0−20) | 0(0−10) |
| I2 | −15(−20−0) | 0(−20−0) | 0 | 6(0−20) | 0(0−10) |
| I3 | 0 | −15(−20−0) | 0(−20−0) | 6(0−20) | 0(0−10) |

Note: The range in parentheses indicates the variety of parameters when controlled.

$$F(x) = \frac{1}{1 + e^{-a(x-\theta)}} - \frac{1}{1 + e^{a\theta}} \tag{11}$$

where the parameter θ is the position of the inflection point of the sigmoid, and $a/4$ is the slope at θ. Here, $\tau$, θ, and $a$ are set as 20, 1.5, and 3. Other parameters of the model varied with each simulation. The simulation specific parameters for *Figure 3* are shown in *Tables 1 and 2* and for *Figure 4* are shown in *Table 3*, respectively.

## Network model with spiking neurons

The BG network with spiking neurons was taken from a previous model by *Chakravarty et al., 2022*. Here, we describe the model briefly and for details we refer the reader to the paper by *Chakravarty et al., 2022*.

### Spiking neuron model

Here, we excluded D1-SPNs because they have a rather small firing rate in PD conditions. The striatal D2-type spiny neurons (D2-SPN), fast-spiking neurons (FSNs) and STN neurons were modelled as standard LIF neurons with conductance-based synapses. The membrane potential $V^x(t)$ of these neurons was given by:

$$C_m^x \frac{dV^x(t)}{dt} = I_e(t) + I_{syn}(t) - g_L^x \left[ V^x(t) - V_{reset}^x \right] \tag{12}$$

where $x \in \{D2 - SPN, FSN, \text{and } STN\}$, $I_e(t)$ is the external current induced by Poisson type spiking inputs (see below), $I_{syn}(t)$ is the total synaptic input (including both excitatory and inhibitory inputs). When $V^x$ reached the threshold potential $V_{th}^x$, the neuron was clamped to $V_{reset}^x$ for a refractory duration $t_{ref}$ = 2ms. All the parameter values and their meaning for D2-SPN, FSN, and STN are summarized in *Appendix 2—tables 2–4*, respectively.

We used the LIF model with exponential adaptation (AdEx) to simulate Proto and Arky neurons of the globus pallidus externa (GPe), with their dynamics defined as

$$C^x \frac{V^x(t)}{dt} = -g_L^x[V^x(t) - V_{reset}^x] - w^x + I_{syn}^x(t) + I_e + g_L^x \Delta_T \exp\left( \frac{V^x(t) - V_T^x}{\Delta_T} \right)$$
$$\tau_w \dot{w}^x = a\left(V^x(t) - V_{reset}^x\right) - w^x$$

where $x \in \{Proto, Arky\}$. Here when $V^x(t)$ reaches the threshold potential ($V_{th}^x$), a spike is generated and $V^x(t)$ as well as $w^x$ will be reset as $V_{reset}^x$, $w^x + b$, respectively, where b denotes the spike-triggered

**Table 2.** Parameters of EII network for *Figure 3*.

| Populations | Synaptic weights | | | External input | Delay |
|---|---|---|---|---|---|
| | E1 | I1 | I2 | | |
| E1 | 0 | 0 | −15(−20−0) | 6 | 0(0−10) |
| I1 | −15(−20−0) | 0(−20−0) | 0 | 6(0−20) | 0(0−10) |
| I2 | 0 | −15(−20−0) | 0(−20−0) | 6(0−20) | 0(0−10) |

Note: The range in parentheses indicates the variety of parameters when controlled.

**Table 3.** Parameters for *Figure 4* (Wilson-Cowan model).

| Populations | Synaptic weights | | | | External input | Delay |
|---|---|---|---|---|---|---|
| | (D2) | (Arky) | (Proto) | (STN) | | |
| (D2) | 0 | −15 | 0 | 0 | 4(0−20) | 2 |
| (Arky) | 0 | 0 | −15 | 15 | 3 | 2 |
| (Proto) | −15 | 0 | −8 | 15 | 1/4 | 2 |
| (STN) | 0 | 0 | −15 | 5 | 4(0−20) | 2 |

The external input to Proto is 1 in *Figure 4c* and 4d while it was changed into 4 in *Figure 4e* to help motif Proto-Arky-D2 oscillate.

adaptation. The parameter values and their meaning for Proto and Arky are specified in *Appendix 2—table 5*. Neurons were connected by static conductance-based synapses. The transient of each incoming synaptic current is given by:

$$I_{\text{syn}}^x(t) = g_{\text{syn}}^x(t)\left[V^x(t) - E_{rev}^x\right]$$

where $x \in \{\text{D2} - \text{SPN, FSN, STN, Arky, and Proto}\}$ is the synaptic reversal potential and $g_{\text{syn}}^x(t)$ is the time course of the conductance transient, given as follows:

$$g_{syn}^x(t) = \begin{cases} J_{syn}^x \dfrac{t}{\tau_{syn}} \exp\left(\dfrac{-\left(t - \tau_{syn}\right)}{\tau_{syn}}\right), & \text{for } t \geq 0 \\ 0, & \text{for } t < 0 \end{cases},$$

where $syn \in \{\text{exc, inh}\}$, $J_{syn}^x$ is the peak of the conductance transient and $\tau_{syn}$ is synaptic time constant. The synaptic parameters are shown in *Appendix 2—table 6*.

Some of model parameters were changed to operate the BG model in specific modes dominated by a 2 or 3 nodes cycle. The *Appendix 2—table 7* and *Appendix 2—table 8* show the parameters of *Figure 5c* and *Figure 5d*, respectively.

## External input

Each neuron in each sub-network of the BG received external input in the form of excitatory Poisson-type spike trains. This input was provided to achieve a physiological level of spiking activity in the network. For more details please see *Chakravarty et al., 2022*. Briefly, the external input was modelled as injection of Poisson spike-train for a brief period of time by using the inhomogeneous_poisson_generator device in NEST. The strength of input stimulation can be controlled by varying the amplitude of the EPSP from the injected spike train.

## STN inhibition experiment

We set a subnetwork of BG to study how STN inhibition affects oscillation when different motifs dominate the system. The connections and external inputs to each neuron in *Figure 5c* and 5d are shown in *Appendix 2—tables 7 and 8*. To simulate the increasing inhibition to STN, the external input to STN was reduced from 1 pA to –99 pA in *Figure 5c* and from 30 pA to –50 pA in *Figure 5d*.

## Data analysis

The estimate of oscillation frequency of the firing rate-based model was done using the power spectral density calculated by pwelch function of MATLAB. The spiking activity of all the neurons in a sub-population were pooled and binned (rectangular bins, bin width = 0.1ms). The spectrum of spiking activity was then calculated for the binned activity using pwelch function of MATLAB.

## Simulation tools

Wilson-Cowan type firing rate-based model was simulated using Matlab. All the relevant differential equations were integrated using Euler method with a time step of 0.01ms. The network of spiking

neurons was simulated in Python 3.0 with the simulator NEST 2.20 *Fardet et al., 2020*. During the simulation, differential equations of BG neurons were integrated using Runga–Kutta method with a time step of 0.1ms.

## Code availability

The code to simulate key results is available at https://github.com/jiezang97/Code-for-Structural-constraints-on-the-emergence-of-oscillations, copy archived at *Zang, 2024*.

## Acknowledgements

We thank Kingshuk Chakravarthy for sharing the code of the basal ganglia network with spiking neurons. We thank Dr. Henri Rihiimaki for helpful comments and suggestions. This work was funded in parts by Swedish Research Council (VR), StratNeuro (to AK), Digital Futures grants (to AK and PH), the Inst. of Advanced Studies, University of Strasbourg, France Fellowship (to AK), and the National Natural Science Foundation of China under Grant No.11572127 and 11872183 (to SL).

## Additional information

### Funding

| Funder | Grant reference number | Author |
|---|---|---|
| Vetenskapsrådet | StratNeuro | Arvind Kumar |
| Vetenskapsrådet | 2018-03118 | Arvind Kumar |
| Digital Futures | | Arvind Kumar |
| National Natural Science Foundation of China-Guangdong Joint Fund | 11572127 | Shenquan Liu |
| National Natural Science Foundation of China | 11872183 | Shenquan Liu |
| Institute of Advanced Studies, University of Strasbourg, France | | Arvind Kumar |

The funders had no role in study design, data collection and interpretation, or the decision to submit the work for publication.

### Author contributions

Jie Zang, Conceptualization, Resources, Software, Formal analysis, Validation, Visualization, Methodology, Writing - original draft, Writing – review and editing; Shenquan Liu, Resources, Writing – review and editing; Pascal Helson, Formal analysis, Writing – review and editing; Arvind Kumar, Conceptualization, Supervision, Funding acquisition, Writing - original draft, Project administration

### Author ORCIDs

Jie Zang ![ORCID] http://orcid.org/0000-0003-2655-3343
Pascal Helson ![ORCID] https://orcid.org/0000-0002-2877-3705
Arvind Kumar ![ORCID] https://orcid.org/0000-0002-8044-9195

Reviewer #1 (Public review): https://doi.org/10.7554/eLife.88777.3.sa1
Reviewer #2 (Public review): https://doi.org/10.7554/eLife.88777.3.sa2
Author response https://doi.org/10.7554/eLife.88777.3.sa3

## Additional files

### Supplementary files
• MDAR checklist

### Data availability
The current manuscript is a computational study, so no data have been generated for this manuscript. Modelling code is uploaded on GitHub. GitHub Link: https://github.com/jiezang97/Code-for-Structural-constraints-on-the-emergence-of-oscillations (copy archived at *Zang, 2024*).

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

## Appendix 1

### Proof of theorems 1 and 2

In this section, we prove theorems 1 and 2 using similar notations as *Curto et al., 2012*.

### Background of fixed point and support

Let $x_0 \in \mathbb{R}^n$ and denote by

$$x^{x_0} : \mathbb{R}^+ \to \mathbb{R}^n$$

$$t \mapsto x^{x_0}(t)$$

a solution of the threshold-linear network dynamical system TLN($W, b$) (*Equation 1*) with $x^{x_0}(0) = x_0$. A fixed point $x^* \in \mathbb{R}^n$ of TLN($W, b$) is defined as a point such that for all $t \geq 0$,

$$\frac{dx^{x^*}(t)}{dt} = 0.$$

Formally, such a point satisfies for all $i \in [n]$,

$$x_i^* = \left[ \sum_{j=1}^n W_{ij} x_j^* + b_i \right]_+ . \tag{13}$$

The support of a fixed point $x^*$,

$$\mathrm{supp}(x^*) \overset{\mathrm{def}}{=} \{ i \in [n], \ x_i^* > 0 \},$$

is the subset of active nodes of $x^*$.

In what follows, we consider a subset $\sigma \subset [n]$ and denote by $\bar{\sigma} = [n] \setminus \sigma$. For a given $n$ by $n$ matrix $W$ and $\sigma, \tilde{\sigma} \subset [n]$, we note

$$\mathrm{W} = \begin{bmatrix} W_{\bar{\sigma}} & W_{\bar{\sigma}\sigma} \\ W_{\sigma\bar{\sigma}} & W_{\sigma} \end{bmatrix}, x = \begin{bmatrix} x_{\bar{\sigma}} \\ x_{\sigma} \end{bmatrix}, b = \begin{bmatrix} b_{\bar{\sigma}} \\ b_{\sigma} \end{bmatrix}.$$

Moreover, we denote by $\left( I_\sigma - W_\sigma; b_\sigma, i \right)$ the matrix $I_\sigma - W_\sigma$ with the $i^{th}$ column replaced by $b_\sigma$.

### Proof of Theorem 1

Theorem 1 says that directed cycles are necessary to help a network to oscillate based on structural conditions.

*Proof of Theorem 1.* If $G$ does not contain any directed cycle, it is called an acyclic digraph. From Proposition 2.1.3 in *Bang-Jensen and Gutin, 2009*, we can always group the nodes into an acycling (or topological) ordering as follows (see *Figure 2a* for an illustration). There exists $k \in [n]$ and a partition $\sigma_1, \sigma_2, \cdots, \sigma_k$ of $[n]$ satisfying, for all $i \in [k]$,

1. $\sigma_i$ has at least one node, and nodes in $\sigma_i$ have no connection between each other,
2. for any node in $\sigma_i$, it can be inhibited and excited only by nodes in $\cup_{j=1}^{i-1} \sigma_j$.

Thus, the nodes of $\sigma_1$ get neither inhibition nor excitation from other nodes, which means their dynamics is independent from the rest of the network. Hence, they all converge exponentially to their unique fixed point, $x_i^* = b_i$ for all $i \in \sigma_1$. Then, once nodes of $\sigma_1$ reached their equilibrium, nodes of $\sigma_2$ will received constant inputs. Hence, nodes of $\sigma_2$ will be stabilized by $\sigma_1$, which means $\sigma_1 \cup \sigma_2$ has a unique globally asymptotically stable fixed point. So on and so forth and thus $\cup_{i=1}^k \sigma_i = [n]$ has a globally asymptotically stable fixed point.

### Proof of Theorem 2

The proof of Theorem 2 follows these lines. We study the fixed point and their stability first when their support is $\sigma = [n]$ and then when $\sigma \subsetneq [n]$ before concluding. However, we first need the following Lemma

## Lemma 1

We consider $x$ a solution of $TLN(W, b)$ under the conditions of Theorem 2 for $W$, $b$ and $x(0)$. Then, for all $t \geq 0$, $x(t)$ admits the same bound as $x(0)$ which is given by *Equation 3*.

*Proof.* Since we have at least one inhibitory node in the network, then $\mathcal{A} \neq \emptyset$. First, for all $k \in [n_I]$, every dynamics of inhibited nodes $a_k$ satisfies

$$-x_{a_k} \leq \frac{dx_{a_k}}{dt} = -x_{a_k} + [-w_{a_k-1}x_{a_k-1} + b_{a_k}]_+ \leq -x_{a_k} + b_{a_k}.$$

Thus defining $\bar{y}_{a_k}$ and $\underline{y}_{a_k}$ as the solutions of

$$\underline{y}_{a_k}(0) = \bar{y}_{a_k}(0) = x_{a_k}(0) \quad \text{and} \quad \frac{d\bar{y}_{a_k}}{dt} = -\bar{y}_{a_k} + b_{a_k}, \qquad \frac{d\underline{y}_{a_k}}{dt} = -\underline{y}_{a_k},$$

we have that for all $t \geq 0$ and $0 \leq x_{a_k}(0) \leq b_{a_k}$,

$$0 \underset{t \to \infty}{\leftarrow} \underline{y}_{a_k}(t) \leq x_{a_k}(t) \leq \bar{y}_{a_k}(t) \underset{t \to \infty}{\to} b_{a_k}.$$

Thus, condition from *Equation 3* implies that for all $t \geq 0$, $0 \leq x_{a_k}(t) \leq b_{a_k}$.

Then, for nodes in between two inhibited nodes, say $j \in \{a_k + 1, \ldots, a_{k-1} - 1\}$, we have by recurrence, for all $t \geq 0$,

$$-x_j \leq \frac{dx_j}{dt} = -x_j + w_{j-1}x_{j-1} \leq -x_j + b_{a_k} \prod_{i=a_k}^{j-1} w_i.$$

Bounding $x_j$ similarly as before we obtain that $0 \leq x_j(0) \leq b_{a_k} \prod_{i=a_k}^{j-1}$ implies that $0 \leq x_j(t) \leq b_{a_k} \prod_{i=a_k}^{j-1} w_i$ for all positive time, which ends the proof.

*Proof of Theorem 2.*

1. When $\sigma = [n]$

First, we consider the fixed point $x^*$ supported by all nodes, $\sigma = [n]$. From Cramer's rule *Cramer, 1750*, we know that if it exists, $x^*$ satisfies

$$x_i^* = \frac{\det\left(I_\sigma - W_\sigma; b_\sigma, i\right)}{\det\left(I_\sigma - W_\sigma\right)} \qquad \text{for } i \in \sigma. \tag{14}$$

We first compute

$$\det(I - W) = \begin{vmatrix} 1 & & & & -W_{1n} \\ -W_{21} & 1 & & & \\ & -W_{32} & 1 & & \\ & & \ddots & \ddots & \\ & & & -W_{nn-1} & 1 \end{vmatrix}$$

$$= \begin{vmatrix} 1 & & & \\ -W_{32} & 1 & & \\ & \ddots & \ddots & \\ & & -W_{nn-1} & 1 \end{vmatrix} + (-1)^n W_{1n} \begin{vmatrix} -W_{21} & 1 & & \\ & -W_{32} & 1 & \\ & & \ddots & 1 \\ & & & -W_{nn-1} \end{vmatrix}$$

$$= 1 + (-1)^{2n-1} \prod_{i=1}^{n} W_{ii-1} = 1 - (-1)^{\sum_{i=1}^{n} \delta_{i,I}} \prod_{i=1}^{n} w_i = 1 - (-1)^{n_I} \prod_{i=1}^{n} w_i.$$

For the numerator, we note that a circular permutation leads to similar results for all the nodes, so we only compute it for the node $i = n$:

$$\det\left(I - W; b, 1\right) = \begin{vmatrix} 1 & & & & b_1 \\ -W_{21} & 1 & & & b_2 \\ & -W_{32} & 1 & & b_3 \\ & & \ddots & \ddots & \vdots \\ & & & -W_{nn-1} & b_n \end{vmatrix} = (-1)^n \sum_{k=1}^{n} (-1)^k b_k \tilde{W}_k,$$

where $\tilde{W}_k = \begin{vmatrix} A_k & \\ & B_k \end{vmatrix} = |A_k| |B_k|$ with $A_1 = \emptyset$, $A_2 = 1$ and for $k \geq 3$,

$$A_k = \begin{vmatrix} 1 & & & \\ -W_{21} & 1 & & \\ & \ddots & \ddots & \\ & & -W_{k-1k-2} & 1 \end{vmatrix} = 1$$

and $B_n = \emptyset$, $B_{n-1} = -W_{nn-1}$ and for $k \leq n - 2$,

$$B_k = \begin{vmatrix} -W_{k+1k} & 1 & \\ & \ddots & 1 \\ & & -W_{nn-1} \end{vmatrix} = (-1)^{n-k} \prod_{i=k+1}^{n} W_{ii-1}.$$

Excited nodes have null external input, $\forall i \in [n] \setminus \mathcal{A}$, $b_i = 0$, so the only terms that will be non-null in $\det\left(I - W; b, 1\right)$ are the one associated to inhibited nodes i.e. nodes in $\mathcal{A}$. With the convention that node $n + 1$ is node 1, we obtain

$$\det(I - W; b,\ 1) = \sum_{k=1}^{n} b_k \prod_{i=k+1}^{n} W_{ii-1} = \sum_{k=1}^{n_I} b_{a_k} \prod_{i=a_k+1}^{n} w_{i-1}(-1)^{\delta_{i-1,I}} = \underbrace{\sum_{k=1}^{n_I} (-1)^{n_I-k} b_{a_k} \prod_{i=a_k}^{n-1} w_i}_{c_k}.$$

We can now conclude on the existence of such a fixed point depending on conditions from *Equation 4* and its inverse *Equation 5*.

If $n_I$ is odd, $\det(I - W) > 0$ and, under both conditions from *Equation 4* and *Equation 5*, $\det\left(I - W; b, i\right) > 0$ for all $i \in [n]$ as $(c_k)_{1 \leq k \leq n_I}$ is an alternating sequence either increasing under *Equation 4* or decreasing under *Equation 5* with both first and last term strictly positive. Thus, $x^*$ defined by *Equation 14* is a fixed point of TLN$(W, b)$ under *Equation 4* and *Equation 5*.

If $n_I$ is even, as *Equation 4* implies *Equation 8* and *Equation 5* implies *Equation 9*,

- *Equation 4* $\Rightarrow \det(I - W) > 0$ and *Equation 5* $\Rightarrow \det(I - W) < 0$,
- under *Equation 4*, $(c_k)_{1 \leq k \leq n_I}$ is an alternating sequence increasing with last term strictly positive so $\det\left(I - W; b, i\right) > 0$,
- under *Equation 5*, $(c_k)_{1 \leq k \leq n_I}$ is an alternating sequence decreasing with first term strictly negative so $\det\left(I - W; b, i\right) < 0$.

Thus, $x^*$ is a fixed point of TLN$(W, b)$ under *Equation 4* or *Equation 5*.

Next we check the stability of this fixed point. To do so, we can use linear theory if there exists a neighborhood of the fixed point in which the system is linear. The dynamics of nodes that are excited are linear so we only have to check that there exists a neighborhood $\mathcal{V}$ of $x^*$ such that for any inhibited node $a \in \mathcal{A}$, the dynamics of node $a$ is linear with respect to other nodes (in our case linear in $a - 1$). We can find such a neighborhood as long as, for any $k \in [n_I]$,

$$\text{either } b_{a_k} < w_{a_k-1} x^*_{a_k-1} \text{ or } b_{a_k} < w_{a_k-1} x^*_{a_k-1}. \tag{15}$$

This is the case since we just showed that the fixed point $x^*$ satisfies $x^*_{a_k} = [b_{a_k} - w_{a_k-1} x^*_{a_k-1}]_+ > 0$ for all $k \in [n_I]$.

We now compute the eigenvalues of $-I + W$, which are the $\lambda$ such that

$$\left|(-I + W) - \lambda I\right| = (\text{-}1\text{-}\lambda)^n + (-1)^{n-1} \prod_{i=1}^{n} w_i (-1)^{\delta_{i,I}} = (-1)^n \left[(1+\lambda)^n - (-1)^{n_I} \prod_{i=1}^{n} w_i\right]$$

is null. Then we have $n$ different solutions that we denote by $\lambda_1, \cdots, \lambda_n$ and such that for all $p \in [n]$,

$$\lambda_p = \begin{cases} \sqrt[n]{\prod_{j=1}^{n} w_j}\, e^{\left(\frac{2p}{n}\right)\pi i} - 1 & \text{if } n_I \text{ is even,} \\ \sqrt[n]{\prod_{j=1}^{n} w_j}\, e^{\left(\frac{2p+1}{n}\right)\pi i} - 1 & \text{if } n_I \text{ is odd.} \end{cases}$$

Based on this, all eigenvalues have negative real parts when $n_I$ is even and $\sqrt[n]{\prod_{i=1}^{n} w_i} < 1$ (**Equation 8**) or when $n_I$ is odd and $\sqrt[n]{\prod_{i=1}^{n} w_i} < \frac{1}{\cos(\pi/n)}$ (**Equation 6**).

In addition, under condition from **Equation 4**, the system has the same linear dynamics for all $t \geq 0$. Indeed, from the bound **Equation 3** on $x(t)$, we have for all $k \in [n_I]$ and $t \geq 0$,

$$w_{a_k-1} x_{a_k-1}(t)$$

Therefore, we obtain that under condition from **Equation 4**, $w_{a_k-1} x_{a_k-1}(t) \leq b_{a_k}$, so all inhibited nodes dynamics have the same linear dynamics for all $t \geq 0$. Moreover, the dynamics of all excited nodes do not change over time and are linear too. Therefore, when **Equation 4** is satisfied, the TLN($W, b$) system is linear and thus, the unique fixed point we found is globally stable (in addition to asymptotically) for both even and odd number of inhibitory nodes.

Hence, under condition from **Equation 4**, the fixed point supported by all nodes is asymptotically stable in loops with both odd or even inhibitory nodes. When $n_I$ is even and **Equation 5** is satisfied, this fixed point is unstable. When $n_I$ is odd and **Equation 5** is satisfied, this fixed point is asymptotically stable under **Equation 6** and unstable otherwise.

2. When $\sigma \subsetneq [n]$

Let us assume that there exists a fixed point $x^*$ with support $\sigma \subsetneq [n]$. We split the following of the proof depending on whether condition given by **Equation 4** or **Equation 5** is satisfied.

## Under condition from *Equation 4*

We can consider a node $p \in \bar{\sigma}$ such that $p - 1 \in \sigma$. Thus $x^*_{p-1} > 0$ and $x^*_p = 0$. If the node $p - 1$ was excitatory, then, as $x^*$ is solution of **Equation 13**, we would also have $x^*_p > 0$. Therefore, node $p - 1$ has to be inhibitory and thus from Lemma 1, we have for all $t \geq 0$,

$$x^*_p(t) = [-w_{p-1} x_{p-1}(t) + b_p]_+ \geq -w_{p-1} b_{p-1} + b_p.$$

Hence, according to the condition stated in **Equation 4**, $x^*_p(t) > 0$ which contradicts the initial assumption $x^*_p = 0$. Thus, under condition from **Equation 4**, the fixed point with $[n]$ is the only possible support.

## Under condition from *Equation 5*

We can consider a node $p \in \sigma$ such that $p - 1 \in \bar{\sigma}$. For the sake of clarity, as we consider cycles, we can say that $p = 1$. Thus $x^*_n = 0$ and $x^*_1 > 0$. If the node $n$ was excitatory, then, as $x^*$ is solution of **Equation 13**, we would also have $x^*_1 = 0$. Therefore, node $n$ has to be inhibitory and thus $x^*_1 = b_1$.

Then we consider the path from node 1 to node $n$. By definition, we know that

$$x^*_j = \begin{cases} w_{j-1} x^*_{j-1} & \text{when node j-1 is excitatory} \\ [b_j - w_{j-1} x^*_{j-1}]_+ & \text{when node j-1 is inhibitory} \end{cases}. \tag{16}$$

Thus, we now compute the possible fixed point such that $x^*_1 = b_1$.

Let us first consider the case in which $n$ is the only inhibitory node. Then, nodes $\{1, \ldots, n-1\}$ are all excitatory. Hence, starting from $x^*_1 = b_1$ and using the first line of **Equation 16**, we obtain that $x^*_n = b_1 \prod_{i=1}^{n-1} w_i > 0$, which contradicts $x^*_n = 0$. Therefore, there is no possible fixed point on $\sigma \subsetneq [n]$ when $n_I = 1$ under **Equation 5**.

Now, assume there are strictly more than one inhibitory node and an odd number. Therefore, $a_2 - 1$ is the next inhibitory node after node $a_1 - 1 = 0$ which if node $n$ when following the cycle. Then we have

$$x_{a_2}^* = [b_{a_2} - b_{a_1} \prod_{i=a_1}^{a_2-1} w_i]_+$$

which is null under condition from **Equation 5**. From the first line of **Equation 16**, all the following excited nodes will have null activity until the next inhibited node. Hence, $x_{a_2}^* = \cdots = x_{a_3-1}^* = 0$ and $x_{a_3}^* = b_{a_3}$ which is then in the same case as node $a_1$. We can thus compute by strong recurrence the possible fixed points. We check their existence by ensuring that the initial condition $x_n^* = 0$ is still satisfied. From the recursive computation, we clearly see that if $x_{a_k-1}^* = 0$, then $x_{a_{k+1}-1}^* = b_{a_k} \prod_{i=a_k}^{a_{k+1}-1} w_i > 0$ (we used the convention given by **Equation 2**) and $x_{a_{k+2}-1}^* = 0$. Therefore, the initial condition $x_n^* = 0$ is only satisfied when the number of inhibitory nodes is even. Hence, the system with odd inhibitory nodes has no fixed point supported by $\sigma \subsetneq [n]$ under condition given by **Equation 5**.

When the number of inhibitory nodes is even, as the recurrence is on two successive inhibited nodes, there are two fixed points depending on the two possible initial conditions: $x_{a_1}^* > 0$ or $x_{a_2}^* > 0$. We denote by $x^{*,1}$ and $x^{*,2}$ these two fixed point having respectively $\sigma_1, \sigma_2 \subsetneq [n]$ as support where

$$\sigma_1 = \bigcup_{k \in [n_I/2]} \{a_{2k+1}, a_{2k+1} + 1..., a_{2k+2} - 1\},$$
$$\sigma_2 = \bigcup_{k \in [n_I/2]} \{a_{2k}, a_{2k} + 1..., a_{2k+1} - 1\}.$$

In particular, $\sigma_1 \cup \sigma_2 = [n]$ and $\sigma_1 \cap \sigma_2 = \emptyset$. Using the convention that $\prod_i^j \cdots = 1$ when $j < i$ and defining the function $\phi : [n] \to \mathcal{A}$ such that $\phi(k) \in \mathcal{A}$ is the last (following the cycle) inhibited node before $k$ (possibly $k$), we can give the exact formula of $x^{*,i}$, $i \in \{1, 2\}$,

$$\begin{cases} x_k^{*,i} = b_{\phi(k)} \prod_{j=\phi(k)}^{k-1} w_j & \text{if } k \in \sigma_i \\ x_k^{*,i} = 0 & \text{if } k \in [n] \setminus \sigma_i. \end{cases}$$

To check the stability, we again use linear theory. To do so, we still have to check the condition given by **Equation 15** (only on inhibited nodes). From the computation of the fixed points, we have shown that under condition from **Equation 5**, for all $a \in \mathcal{A}$, either $x_a^* = 0$ because $b_a < w_{a-1}x_{a-1}^*$ or $x_a^* > 0$ because $b_a > w_{a-1}x_{a-1}^*$. Thus condition given by **Equation 15** is satisfied. Hence, we now compute the eigenvalues of $-I_{\sigma_i} + W_{\sigma_i}$ for $i \in \{1, 2\}$, which are the solutions of

$$\left| (-I_{\sigma_i} + W_{\sigma_i}) - \lambda I_{\sigma_i} \right| = 0.$$

To do so we write the weight matrix $W_{\sigma_i}$.
Both $W_{\sigma_1}$ and $W_{\sigma_2}$ are similar so we only write

$$W_{\sigma_1} = \begin{vmatrix} C_1 & & \\ & \ddots & \\ & & C_{n_I} \end{vmatrix} \quad \text{with} \quad C_k = \begin{vmatrix} 0 & & & \\ W_{a_{2k}} & 0 & & \\ & & \ddots & \\ & & W_{a_{2k+1}-1 a_{2k+1}-2} & 0 \end{vmatrix}.$$

We deduce that

$$|(-I_{\sigma_i} + W_{\sigma_i}) - \lambda I_{\sigma_i}| = (-1 - \lambda)^{\text{card}(\sigma_i)},$$

so $\lambda = -1$ and both fixed points are asymptotically stable.

Eventually, when **Equation 4** is satisfied, the system with even or odd inhibitory nodes has only one globally asymptotically stable fixed point on $[n]$. When $n_I$ is even and **Equation 5** is satisfied, the system has only two asymptotically stable fixed points on $\sigma \subsetneq [n]$ and one unstable fixed point on $[n]$. When $n_I$ is odd, the system has an asymptotically stable fixed point on $[n]$ under conditions

*Equations 5 and 6*; otherwise, it has a unique unstable fixed point on $[n]$ (thus no stable fixed point) under conditions *Equations 5 and 7*.

# Appendix 2

## Supplementary information

**Appendix 2—table 1.** Parameters for the EI network in *Figure 3* (Wilson-Cowan model).

| Populations | Synaptic Weights (E) | Synaptic Weights (I) | External Input | Delay |
|---|---|---|---|---|
| E | 10 (0–20) | –15 (–20–0) | 6 (0–20) | 2 (0–10) |
| I | 15 (0–20) | –10 (–20–0) | 0 | 2 (0–10) |

**Appendix 2—table 2.** Parameters of D2-SPN neurons (LIF model with conductance-based synapses).

| Name | Value | Description |
|---|---|---|
| $V_{reset}$ | –85.4 mV | Reset value for $V_m$ after a spike |
| $V_{th}$ | –45 mV | Spike threshold |
| $\tau_{syn}^{ex}$ | 0.3ms | Rise time of excitatory synaptic conductance |
| $\tau_{syn}^{in}$ | 2ms | Rise time of inhibitory synaptic conductance |
| $E_L$ | –85.4 mV | Leak reversal potential |
| $E_{ex}$ | 0 mV | Excitatory reversal potential |
| $E_{in}$ | –64 mV | Inhibitory reversal potential |
| $I_e$ | 0 pA | External input current |
| $C_m$ | 157 pF | Membrane capacitance |
| $g_L$ | 6.46 nS | Leak conductance |
| $t_{ref}$ | 2ms | Duration of the refractory period |

**Appendix 2—table 3.** Parameters of FSN neurons (LIF model with conductance-based synapses).

| Name | Value | Description |
|---|---|---|
| $V_{reset}$ | –65 mV | Reset value for $V_m$ after a spike |
| $V_{th}$ | –54 mV | Spike threshold |
| $\tau_{syn}^{ex}$ | 0.3ms | Rise time of excitatory synaptic conductance |
| $\tau_{syn}^{in}$ | 2ms | Rise time of inhibitory synaptic conductance |
| $E_L$ | –65 mV | Leak reversal potential |
| $E_{ex}$ | 0 mV | Excitatory reversal potential |
| $E_{in}$ | –76 mV | Inhibitory reversal potential |
| $I_e$ | 0 pA | External input current |
| $C_m$ | 700 pF | Membrane capacitance |
| $g_L$ | 16.67 nS | Leak conductance |
| $t_{ref}$ | 2ms | Duration of the refractory period |

**Appendix 2—table 4.** Parameters of STN neurons (LIF model with conductance-based synapses).

| Name | Value | Description |
|---|---|---|
| $V_{reset}$ | –70 mV | Reset value for $V_m$ after a spike |

*Appendix 2—table 4 Continued on next page*

*Appendix 2—table 4 Continued*

| Name | Value | Description |
|------|-------|-------------|
| $V_{th}$ | –64 mV | Spike threshold |
| $\tau_{syn}^{ex}$ | 0.33ms | Rise time of excitatory synaptic conductance |
| $\tau_{syn}^{in}$ | 1.5ms | Rise time of inhibitory synaptic conductance |
| $E_L$ | –80.2 mV | Leak reversal potential |
| $E_{ex}$ | –10 mV | Excitatory reversal potential |
| $E_{in}$ | –84 mV | Inhibitory reversal potential |
| $I_e$ | 1 pA | External input current |
| $C_m$ | 60 pF | Membrane capacitance |
| $g_L$ | 10 nS | Leak conductance |
| $t_{ref}$ | 2ms | Duration of the refractory period |

**Appendix 2—table 5.** Parameters of Proto and Arky neurons (LIF model with AdEx).

| Name | Proto | Arky | Description |
|------|-------|------|-------------|
| a | 2.5 nS | 2.5 nS | Subthresholded adaptation |
| b | 105 pA | 70 pA | Spike-triggered adaptation |
| $\Delta_T$ | 2.55ms | 1.7ms | Slope factor |
| $\tau_w$ | 20ms | 20ms | Adaptation time constant |
| $V_{reset}$ | –60 mV | –60 mV | Reset value for $V_m$ after a spike |
| $V_{th}$ | –54.7 mV | –54.7 mV | Spike threshold |
| $\tau_{syn}^{ex}$ | 1 ms | 4.8ms | Rise time of excitatory synaptic conductance |
| $\tau_{syn}^{in}$ | 5.5ms | 1 ms | Rise time of inhibitory synaptic conductance |
| $E_L$ | –55.1 mV | –55.1 mV | Leak reversal potential |
| $E_{ex}$ | 0 mV | 0 mV | Excitatory reversal potential |
| $E_{in}$ | –65 mV | –65 mV | Inhibitory reversal potential |
| $I_e$ | 1 pA | 12 pA | Constant input current |
| $C_m$ | 60 pF | 40 pF | Membrane capacitance |
| $g_L$ | 1 nS | 1 nS | Leak conductance |
| $t_{ref}$ | 2ms | 2ms | Duration of the refractory period |

**Appendix 2—table 6.** Synaptic conductance weight and delay parameters in LIF model.

| Synapse | Value (nS) | Delay | Value (ms) |
|---------|-----------|-------|-----------|
| $J_{D2}^{D2}$ | -0.35 | $\Delta_{D2}^{D2}$ | 1.7 |
| $J_{D2}^{FSN}$ | -2.6 nS | $\Delta_{D2}^{FSN}$ | 1.7 |
| $J_{D2}^{Arky}$ | -0.04 nS | $\Delta_{D2}^{Arky}$ | 7 |
| $J_{FSN}^{FSN}$ | -0.4 nS | $\Delta_{FSN}^{FSN}$ | 1.7 |
| $J_{FSN}^{Arky}$ | -0.25 nS | $\Delta_{FSN}^{Arky}$ | 7 |

*Appendix 2—table 6 Continued on next page*

*Appendix 2—table 6 Continued*

| Synapse | Value (nS) | Delay | Value (ms) |
|---|---|---|---|
| $J^{Proto}_{FSN}$ | -1 nS | $\Delta^{Proto}_{FSN}$ | 7 |
| $J^{Proto}_{Proto}$ | -1.3 nS | $\Delta^{Proto}_{Proto}$ | 1 |
| $J^{D2}_{Proto}$ | -1.08 nS | $\Delta^{D2}_{Proto}$ | 7 |
| $J^{STN}_{Proto}$ | 0.175 nS | $\Delta^{STN}_{Proto}$ | 2 |
| $J^{Arky}_{Arky}$ | -0.11 nS | $\Delta^{Arky}_{Arky}$ | 1 |
| $J^{Proto}_{Arky}$ | -0.35 nS | $\Delta^{Proto}_{Arky}$ | 1 |
| $J^{STN}_{Arky}$ | 0.24 nS | $\Delta^{STN}_{Arky}$ | 2 |
| $J^{Proto}_{STN}$ | -0.3 nS | $\Delta^{Proto}_{STN}$ | 1 |

**Appendix 2—table 7.** Number of connections on each neuron and constant input current for *Figure 5c* (LIF model).

| Populations | D2 | Arky | Proto | STN | Constant input current (pA) |
|---|---|---|---|---|---|
| D2 | 504 | 100 | 0 | 0 | 0 |
| Arky | 0 | 5 | 50 | 30 | 50 |
| Proto | 500 | 0 | 25 | 30 | 50 |
| STN | 0 | 0 | 30 | 0 | 1/–49/–99 |

Note: To simulate the increasing inhibition to STN, the constant input current to STN was changed from 1 pA to –49 pA and then to –99 pA.

**Appendix 2—table 8.** Number of connections on each neuron and constant input current for *Figure 5d* (LIF model).

| Populations | D2 | Arky | Proto | STN | Constant input current (pA) |
|---|---|---|---|---|---|
| D2 | 504 | 10 | 0 | 0 | 0 |
| Arky | 0 | 5 | 25 | 30 | 1 |
| Proto | 500 | 0 | 25 | 150 | –10 |
| STN | 0 | 0 | 150 | 0 | 30/–10/–80 |

Note: To simulate the increasing inhibition to STN, the constant input current to STN was changed from 30 pA to –10 pA and then to –50 pA.

