## [Editor Report · eLife assessment]

The present study offers **valuable** insights into the emergence of oscillations in neural networks. It underscores the importance of achieving a delicate balance between excitatory and inhibitory links, and deals with the topological conditions for oscillations. The study provides **solid** evidence in simple networks based on formal mathematical theory and advanced simulations, but the wider implications to biological networks would require a more detailed investigation into delays and nonlinearities.

---

## [Referee Report · Reviewer #1 (Public review)]

Summary:

Authors study appearance of oscillations in motifs of linear threshold systems, coupled in specific topologies. They derive analytically conditions for appearance of oscillations, in the context of excitatory and inhibitory links. They also emphasize the higher importance of the topology, compared to the strength of the links, though it is not straightforward to apply this for brain networks where the weights can be distributed several orders of magnitude. Finally the results are confirmed with WC oscillators. The findings are to some extent confirmed with spiking neurons, though here results are less clear.

Overall, the results are sound from a theoretical perspective, but I still find hard to believe that they are of significant relevance for biological networks, or in particular for the oscillations of BG-thalamus-cortex loop in PD. I find motifs in general to be too simplistic for multiscale and generally large networks as it is the case in the brain. Moreover, the division on regions is more or less arbitrary by definition, and having such a strong dependence on odd/even number of inhibitory links is far from reality. Another limitation is the fact that the cortex is considered as a single node. Similarly, decomposing even such a coarse network in all possible (238 in this case) motifs doesn't seem of much relevance, when I'd assume that the emergence of pathological rhythms is more of an emergent phenomena.

Strengths:

From the point of nonlinear dynamics, the results are solid, and the intuition behind the proofs of the theorems is well explained.

Weaknesses:

As stated in the summary, I find the work to be too theoretical without a real application for the brain dynamics, where the networks are generally very large. The odd/even number rule is too strict, and talking about fixed and definite number of cycles in actual brain seems too simplistic. Moreover, the cortex is considered as a single node, and finally the impact of the delays is ignored even though they define the synchronizability of the brain network, and previous works on the amplitude reduction due to the time-delays in difference-coupled networks of oscillators is not mentioned.

---

## [Referee Report · Reviewer #2 (Public review)]

The authors present here a mathematical and computational study of the topological/graph theory requirements to obtain sustained oscillations in neural network models. A first approach mathematically demonstrates that, for a given network of interconnected neural populations (understood in the sense of dynamical systems) requires an odd number of inhibitory populations to sustain oscillations. The authors extend this result via numerical simulations of (i) a simplified set of Wilson-Cowan networks, (ii) a simplified circuit of the cortico-basal ganglia network, and (iii) a more complex, spike-based neural network of basal ganglia network, which provides insight on experimental findings regarding abnormal synchrony levels in Parkinson's Disease (PD).

The work elegantly and effectively combines a solid mathematical proof with careful numerical simulations at different levels of description, which is uncommon and provides additional layers of confidence to the study. Furthermore, the authors included detailed sections to provide intuition about the mathematical proof, which will be helpful for readers less inclined to the perusal of mathematical derivations. Its insightful and well-informed connection with a practical neuroscience problem, the presence of strong beta rhythms in PD, elevates the potential influence of the study and provides testable predictions.

In its updated form, the authors have solved the most pressing issues of the study, by acknowledging the limitations of their work regarding the effects of delays in oscillations, and addressing some of these effects in new simulations. Although some interesting simulations are still not presented in the revised version, they could constitute the focus of future work to complement the conclusions presented here. The absence of explanations for some of the figures and panels has been corrected, and the issues with grammar and lack of clarity have been improved. This important work is therefore now improved.

---

## [Author Response]

The following is the authors’ response to the original reviews.

**Public Reviews:**

**Reviewer #1 (Public Review):**
Summary:The authors study the appearance of oscillations in motifs of linear threshold systems, coupled in specific topologies. They derive analytical conditions for the appearance of oscillations, in the context of excitatory and inhibitory links. They also emphasize the higher importance of the topology, compared to the strength of the links. Finally, the results are confirmed with WC oscillators, which are also linear. The findings are to some extent confirmed with spiking neurons, though here results are less clear, and they are not even mentioned in the Discussion.Overall, the results are sound from a theoretical perspective, but I still find it hard to believe that they are of significant relevance for biological networks, or in particular for the oscillations of BG-thalamus-cortex loop in PD. I find motifs in general to be too simplistic for multiscale and generally large networks as is the case in the brain. Moreover, the division of regions is more or less arbitrary by definition, and having such a strong dependence on an odd/even number of inhibitory links is far from reality. Another limitation is the fact that the cortex is considered a single node. Similarly, decomposing even such a coarse network in all possible (238 in this case) motifs doesn't seem of much relevance, when I assume that the emergence of pathological rhythms is more of an emergent phenomenon.Strengths:From the point of view of nonlinear dynamics, the results are solid, and the intuition behind the proofs of the theorems is well explained.Weaknesses:As stated in the summary, I find the work to be too theoretical without a real application in biological systems or the brain, where the networks are generally very large.

We respectfully disagree with the reviewer here. The second half of the paper is all about explaining a biological problem. We have shown the validity of our theoretical results (which indeed were obtained in idealized settings) to explain emergence of oscillations in the basal ganglia. We clearly show that our theoretical results hold both in a rate-based model and in a network model with spiking neurons. The model with spiking neurons is one of the most complete network models of the basal ganglia available in the literature. So we emphasize that we have provided a clear application of our results for the brain networks.

It is not the problem in the simplicity of the model or of the topology, it is often the case that the phenomena are explained by very reduced systems, but the problem is that the applicability of the finding cannot be extended. E.g. the Kuramoto model uses all-to-all coupling, or similar with QIF neurons which also need to follow a Lorentzian distribution in order to derive a mean field.

We do not understand this comment. There is no need to extend these results to a network of Kuramoto models because in that setting we already assume that individual nodes/populations are oscillating – there is no problem of emergence of oscillations. Here, we are specifically considering a setting in which nodes themselves are not oscillators. We agree that we, at this point, have no insight into how to extend our analytical proof to a situation where individual nodes are spiking.

But in those cases, relaxing the strict conditions that were necessary for the derivations, still conserves the main findings of the analysis, which I don't see being the case here. The odd/even number rule is too strict, and talking about a fixed and definite number of cycles in the actual brain seems too simplistic.

We have clearly relaxed most of our assumptions when we considered a network model of basal ganglia in which each subpopulation is a collection of spiking neurons. And as we have shown our results still hold (see Figure 5). Again our model is about oscillations in a network of networks i.e. network of brain regions.

At meso-scale it is not unreasonable to find such cycles and even-odd number rules. We have shown this for the case of a cortico-basal ganglia model. We can also extend this to cortico-thalamic networks and so on. We have already emphasized this point in the introduction to avoid any confusion: see lines 62-66 – “We prove this conjecture for the threshold-linear network (TLN) model without delays which can closely capture the dynamics of neural populations. Therefore, it is implicit that our results do not hold at the neuronal level but rather at the level of neuron populations/brain regions e.g. the basal ganglia (BG) network which can be described a network of different nuclei.” and lines 69-70 – ’Within the framework of the odd-cycle theory, distinct nuclei are associated with either excitatory or inhibitory nodes.’

Being linear is another strong assumption, and it is not clear how much of the results are preserved for spiking neurons, even though there is such an analysis, or maybe for other nonlinear types of neuronal masses.

Clearly our results hold in a network of spiking neurons (see Figure 5). It is of course interesting to ask whether our results hold in a network where individual spiking neurons have more complex spiking behavior like AdEx or Quadratic IF. But that kind of analysis deserves a full manuscript on its own.

Delays are also mentioned, and their impact on the oscillatory networks is as expected: it reduces the amplitude, but there is no link to the literature, where this is an established phenomenon during synchronization. Finally, the authors should also discuss the time-delays as a known phenomenon to cause or amplify oscillations at different frequencies in a network of coupled oscillators, e.g Petkoski & Jirsa Network Neuroscience 2022, Tewarie et al. NeuroImage 2019, Davis et al. Nat Commun 2021.

This is indeed a weakness of our model. But as the reviewer already knows, dynamical systems with delays are very difficult to analyze analytically. We have mentioned this in the limitations of the model and the analysis. In our simulations we have considered delays and when the delays are within reasonable limits our results hold.

**Reviewer #2 (Public Review):**

**Summary:**
The authors present here a mathematical and computational study of the topological/graph theory requirements to obtain sustained oscillations in neural network models. A first approach mathematically demonstrates that a given network of interconnected neural populations (understood in the sense of dynamical systems) requires an odd number of inhibitory populations to sustain oscillations. The authors extend this result via numerical simulations of (i) a simplified set of Wilson-Cowan networks, (ii) a simplified circuit of the cortico-basal ganglia network, and (iii) a more complex, spike-based neural network of basal ganglia network, which provides insight on experimental findings regarding abnormal synchrony levels in Parkinson's Disease (PD).Strengths:The work elegantly and effectively combines solid mathematical proof with careful numerical simulations at different levels of description, which is uncommon and provides additional layers of confidence to the study. Furthermore, the authors included detailed sections to provide intuition about the mathematical proof, which will be helpful for readers less inclined to the perusal of mathematical derivations. Its insightful and well-informed connection with a practical neuroscience problem, the presence of strong beta rhythms in PD, elevates the potential influence of the study and provides testable predictions.Weaknesses:In its current form, the study lacks a more careful consideration of the role of delays in the emergence of oscillations. Although they are addressed at certain points during the second part of the study, there are sections in which this could have been done more carefully, perhaps with additional simulations to solidify the authors' claims. Furthermore, there are several results reported in the main figures which are not explained in the main text. From what I can infer, these are interesting and relevant results and should be covered. Finally, the text would significantly benefit from a revision of the grammar, to improve the general readability at certain sections. I consider that all these issues are solvable and this would make the study more complete.

This point has been made by the first reviewer as well. So we repeat our answer:

This is indeed a weakness of our model. But as the reviewer already knows, dynamical systems with delays are very difficult to analyze analytically. We have mentioned this in the limitations of the model and the analysis. In our simulations we have considered delays and when the delays are within reasonable limits our results hold.

**Reviewer #2 (Recommendations For The Authors):**
As mentioned in my comments above, I think that the work is already quite solid and relevant but would significantly improve if some issues were addressed:

We would like to thank the reviewer for valuable comments and constructive feedback which has helped us greatly improve the manuscript.

1. While the authors acknowledge early on the limitations of this study in terms of not considering plasticity or neuron biophysics (line 72), I think that the absence of propagation delays should be explicitly included here. This absence leads to inaccuracies --for example, the sentence "Consider a small network of two nodes. If we connect them mutually with excitatory synapses, intuitively we can say that the two-population network will not oscillate" (line 74) is only correct if the delays (or signal latencies) are zero. With a proper delay, two excitatory neurons can engage in oscillations with a period given by two times the value of the delay.A similar situation happens for inhibitory neurons, where the winner-take-all dynamics described in line 77 is only valid for zero delay. It is known that a homogeneous population of inhibitory spiking neurons with delayed synapses can lead to fast oscillations (Brunel and Hakim 1999), something which is also valid for the equivalent inhibitory single node with delayed self-inhibition. Indeed, a circuit of two inhibitory populations with delayed self- and cross-inhibition can generate oscillations, contradicting the main conclusion of the odd number of inhibitory nodes needed for oscillations.Because of these considerations, I think the authors should be more careful when explaining the effects of delays, and state that their main results on the link between oscillations and having an odd number of inhibitory nodes are not valid when delays are considered. They could modify the sentences in lines 72-77 above and include a supplementary figure right after their simulation study for the Wilson-Cowan (to explain the examples above, and also the one in the next point).

The reviewer has brought up a critical point regarding the impact of propagation delays, and we completely concur with your assessment. In our study, we indeed did not comprehensively consider the effects of propagation delays in cycles with even inhibition, which may introduce inaccuracies in our conclusions.

We note that in the Wilson-Cowan model with delays, certain cycles with even number of inhibitory links can also generate oscillations with a period equal to twice the delay value. However, in our hand such oscillations were transient and dissipated quickly.

To better reflect the limitations of our research, we have made significant modifications to the relevant sections in our manuscript.

In line 100, we've added text to explicitly state that we considered delays in our simulations and acknowledged their potential to generate oscillations ("Given the importance of delays in biological network such as BG, we will consider them in the simulations.").

In line 102, we've clarified that our conclusions are based on a scenario without delays ("In this following, we give simple examples of the possibility of oscillation (or not) based on the connectivity characteristics of small networks without delays. Let us start with a network of two nodes.").

Additionally, in line 230, we've included a reference figure supplement 3-2 to highlight the outcomes in terms of oscillations ("EII network only resulted in transient oscillations (Fig. 3, figure supplement 3-1, figure supplement 3-2)").

In lines 234-237, we've added a sentence discussing the role of synaptic delays in generating transient oscillations in cycles with an even number of inhibitory components, referring to figure supplement 3-2 ("In networks with even number of inhibitory connections (e.g. EII, EEE, II), synaptic delays are the sole mechanism for initiating oscillations, however, unless delays are precisely tuned such oscillations will remain transient (see Supplementary figure supplement 3-2)").

Moreover, in response to the reviewer’s suggestion, we have included an additional figure supplement 3-2 to illustrate how cycles with even inhibitory components generate transient oscillations when propagation delays are taken into account. This figure provides a visual representation of the phenomenon and enhances the clarity of our findings.

1. In Figure 3, two motifs (III and EII) are explored to demonstrate the validity of the results across different parameters. Delays don't seem to play a disruptive role in these two cases, but the results seem to be different for other motifs not considered here. Aside from the examples mentioned above, I can imagine how a motif of EEE (i.e. a circle of three excitatory Wilson-Cowan neurons) would display oscillations when delays are included, as the activation would 'circulate' along the ring. However, this EEE motif has an even number of inhibitory units (or perhaps zero is considered an exception, but if so it's not mentioned in the text).

We thank the reviewer for this observation regarding Figure 3. Indeed, the impact of delays may differ for other motifs not considered in our study. For example, as the reviewer has correctly anticipated, a motif of EEE (a circular network of three excitatory Wilson-Cowan neurons) would exhibit oscillations when delays are included, as activation could 'circulate' along the ring.

To address this concern,we have performed new simulations (added as a new supplementary figure supplement 3-2). As illustrated in figure supplement 3-2, oscillations may indeed arise in the EEE motif when delays are introduced. However, these oscillations will eventually dissipate – at least with our settings.

1. Figures 1b, 1c, and 4e display interesting results, but these are absent from the main text. Please include the description of those results. Particularly the case of Figs 1b and 1c seems very relevant to understanding the main results in the context of more complex networks, in which multiple loops with odd and even numbers of inhibitory units would coexist in the network. Does the number of odd-inhibitory loops in a given network affect somehow the power or frequency of the resulting network oscillations? It would be interesting to show this.

Indeed, we did not explain Figs 1b,c and 4e properly. Now we have revised the manuscript in the following way to incorporate these results:

In lines 124-128, we added the following text to introduce the concept: "We can generalize these results to cycles of any size, categorizing them into two types based on the count of their inhibitory connections in one direction (referred to as the odd cycle rule, as illustrated in Fig. 1b). More complex networks can also be decomposed into cycles of size 2…N (where N is number of nodes), and predict the ability of the network to oscillate (as shown in Fig. 1c)"In line 298, we included the following text to highlight the relevant result: "Next, we removed the STN output (equivalent to inhibition of STN), the Proto-D2-Arky subnetwork generated oscillations for weak positive inputs to the D2-SPNs (Fig.4e, bottom)."

How the number of odd/even loops affect the frequency is an interesting question. Intuitively there should be a relation between the two. However, a complete treatment of this question is beyond the scope of the manuscript but we think that in a network with identical node properties, more odd cycles should imply higher oscillation power.

1. The cortico-BG model is focused on how inactivating STN could suppress (or not) beta oscillations, following experimental observations. However, besides mechanisms for extinguishing oscillations, it would be interesting to see if the progressive emergence of pathological beta oscillations could be explained by the modification of some of the nodes in the model (for example, explicitly mimicking the loss of dopaminergic neurons in the substantia nigra). This could be a very interesting additional figure in the main text.

This is an interesting suggestion. Something similar has been already done – e.g. Kumar et al. (2010) showed that progressive increase of inhibition of GPe can lead to oscillations. Similarly Holgado et al. (2008) showed how progressive change in the mutual connectivity between STN and GPe can cause oscillations. More recently, Ortone et al. (PloS Comp. Biol 2023) and Azizpour et al. (2023 Bioarxiv) have also shown the effect of progressive change in individual node properties on oscillations in basal ganglia using numerical simulations. Our work in a way provides the theoretical backing to their work. Therefore, we think it is not necessary to again show these results in our model. Instead we have cited these papers. Lines 392-396

1. I observed some grammatical inconsistencies in the text, some of them are indicated below. I would suggest carefully going through the text to correct those issues or seeking help with editing.-line 32 "...which can closely capture the neural population dynamics". Which population dynamics? Do the authors refer to general neural dynamics?-line 33 "long term behavior" -> long-term behavior-line 68 "given the ionic channel composition" -> "given its ionic channel composition"

We apologize for the grammatical inconsistencies in our manuscript. We have made the necessary corrections to improve the clarity and accuracy of our text.

**Reviewer #3 (Recommendations For The Authors):**
This manuscript is useful for analytically showing that a cyclic network of threshold-linear neural populations can only oscillate if it has an odd number of inhibitory nodes with strong enough connections. Establishing this result, which holds under rather narrow assumptions, relies on standard tools from dynamical system theory. I find the strength of support for this result to be incomplete for the reasons detailed below:Although the mathematical arguments used appear to be correct, the manuscript lacks in rigor and clarity. For instance, the main result presented in theorem 2 is stated in a very unclear fashion: aside from the oddity of the number of inhibitory nodes, there are two conditions to check, which determines four cases. This can be explained in a much more straightforward way without introducing four relations in equations 4-7.

We acknowledge the reviewer’s concern regarding the presentation of the main result in Theorem 2.

We would like to emphasize that the introduction of four relations in equations 4-7 was intended to provide a detailed and transparent exposition of the conditions for the main result. While we understand that this approach may appear less straightforward, it allows for a more comprehensive understanding of the underlying logic and the multiple factors influencing the outcomes.

However, we are open to suggestions for more concise and clear ways to express these conditions if the reviewer has specific recommendations or if there are alternative approaches that the reviewer believes would be more effective in conveying the information.

Moreover, equation 3 in that same theorem is clearly wrong.

We sincerely apologize for the typographical error in equation 3 within the same theorem. We thank the reviewer for noticing this. We have revised the text to rectify this mistake. The equation has now been corrected to ensure its accuracy.

The proof of theorem 2 relies on standard linear algebra and can be improved as well: there are typos, approximations, and missing words (see line 664). The rigor of the exposition is also unsatisfactory. For instance, the proof of Lemma 1 ends with the sentence: "Similarly as before, the convergence of the dynamics driven by the left and right terms ends the proof". I don't know what this means.

We thank the reviewer for the comments and suggestions. We have made the necessary adjustments to enhance the rigor and clarity of our mathematical reasoning in the revised manuscript.

In line 644, we have provided clarification for the sentence you found unclear. The revised version now offers a more precise explanation that should help in understanding the proof.

At the same time, the intuitive arguments presented in the main text are vague at best and do not really help grasping the possible generalizability of the results. For instance, I do not understand the message of panel B in Figure 2 and there seems to be no explanation about it in the main text.

The main purpose of Figure 2B is to offer a visual representation of the concept and to serve as an aid for readers who may prefer a graphical illustration over extensive equations. While we understand that the figure may not provide a complete explanation on its own, it is intended to complement the text and mathematical content presented in the main text. In the revised version we have added the explanation of Figure 2B.

Aside from the analytical result, most of the paper consists in simulating networks with distinct inhibitory cyclic structure to validate the theoretical argument. I do not find this approach particularly convincing due to the qualitative nature of the numerical results presented. There is little quantitative analysis of the network structure in relation to the emergence of oscillations. It is also hard to judge whether the examples discussed are cherry picked or truly representative of a large class of dynamics.

The reviewer has a valid concern about numerical simulations and qualitative nature of the results. We would like to provide some perspective on our approach.

In our paper, the primary focus is on the mathematical proof, which rigorously establishes the existence of our results. However, we understand that numerical simulations are valuable for illustrating the applicability of the theoretical framework and providing insights into the practical implications.

If we get into the quantitative description of all the results, the manuscript will become prohibitively long. We acknowledge that there is a balance to be struck between theory and numerical examples in a research paper. We believe that, in conjunction with the mathematical proof, the numerical simulations serve the purpose of illustrating the existence of our results in specific examples. While we cannot provide an exhaustive exploration of all possible network structures, we have chosen representative cases to demonstrate the applicability of our findings. Some of these are already provided in figure supplements S3-1 and S3-3. In the absence of specific suggestions from the reviewer we would like to leave it as is.

Moreover, the authors apply their cycle analysis to real-world networks by considering cycles of inhibitory nodes independently, whereas the same nodes can belong to several cycles. I find it hard to believe that considering these cycles independently should be enough to make predictions about the emergence of oscillations, as these cycles must interact with one another via shared nodes. I do not understand the color coding used to mark distinct cycles in supplementary figures. There is also not enough information to understand figures in the main text. For instance, I do not understand what the grids are representing in panel B, Figure 4.

We have clarified the color coding and added more information to understand the figures.We appreciate the reviewer’s concern about our application of cycle analysis to real-world networks and the clarity of our figures. It is not a matter of belief – we have provided a mathematical proof and complemented that with illustrative examples from real-world networks i.e. cortico-basal ganglia network with both rate-based and spiking neurons. Clearly our results hold.

Regarding the color coding in supplementary figures, we have revised the color scheme to make it more intuitive and informative in caption of figure 4: we use different colors to mark potential oscillators in each motif in BG, and each color means an oscillator from panel a. For more details, see figure supplements 4-1–4-6. The colors now represent distinct cycles more clearly, helping readers better interpret the figures.